

# Towards quantifying information flows: Relative entropy in deep neural networks and the renormalization group

**Johanna Erdmenger[1], Kevin T. Grosvenor[2] and Ro Jefferson[3]**

**1** Institute for Theoretical Physics and Astrophysics and Würzburg-Dresden
Cluster of Excellence ct.qmat, Julius-Maximilians-Universität Würzburg,
Am Hubland, 97074 Würzburg, Germany
**2** Max Planck Institute for the Physics of Complex Systems and Würzburg-Dresden
Cluster of Excellence ct.qmat, Nöthnitzer Str. 38, 01187 Dresden, Germany
**3** Nordita, KTH Royal Institute of Technology and Stockholm University,
Hannes Alfvéns väg 12, SE-106 91 Stockholm, Sweden

## Abstract

We investigate the analogy between the renormalization group (RG) and deep neural networks, wherein subsequent layers of neurons are analogous to successive steps along the RG. In particular, we quantify the flow of information by explicitly computing the relative entropy or Kullback-Leibler divergence in both the one- and two-dimensional Ising models under decimation RG, as well as in a feedforward neural network as a function of depth. We observe qualitatively identical behavior characterized by the monotonic increase to a parameter-dependent asymptotic value. On the quantum field theory side, the monotonic increase confirms the connection between the relative entropy and the $c$-theorem. For the neural networks, the asymptotic behavior may have implications for various information maximization methods in machine learning, as well as for disentangling compactness and generalizability. Furthermore, while both the two-dimensional Ising model and the random neural networks we consider exhibit non-trivial critical points, the relative entropy appears insensitive to the phase structure of either system. In this sense, more refined probes are required in order to fully elucidate the flow of information in these models.

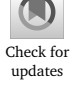

# 1   Introduction

In recent years, a number of works have pointed to similarities between deep neural networks and the renormalization group (RG) [1–10]. This connection was originally made in the context of lattice models, where decimation RG bears a superficial resemblance to certain feedforward neural network architectures. Structurally, both systems involve a coarse-graining procedure that extracts relevant information by marginalizing over hidden degrees of freedom. In the Wilsonian approach to RG, ultra-violet (UV) degrees of freedom are integrated out above some energy scale, resulting in an effective field theory that allows to make accurate predictions about the infra-red (IR). In Bayesian language, this simply corresponds to marginalizing over the high-energy information that low-energy observers cannot access. Similarly, the idea behind deep neural networks is the extraction of increasingly abstract[1] features from the input data—e.g., characterizing the equivalence class of "cats" from a stream of pixel values.

One attempt to formalize this connection was made in [2], which proposed an exact mapping between variational RG and restricted Boltzmann machines (RBMs) [13]. The latter belong to a class of deep neural networks known as energy-based models, and are so-named because the stationary distribution for the state of the network $\mathbf{x} = \{x_0, \ldots, x_n\}$, where $x_i$ is the state (e.g., on or off) of the $i^{\text{th}}$ neuron, is of the canonical form $p(\mathbf{x}) = Z^{-1}e^{-H(\mathbf{x})}$, where $H(\mathbf{x})$ is the Hamiltonian and $Z = \text{tr}_{\mathbf{x}} e^{-H(\mathbf{x})}$ is the partition function. The Hamiltonian will depend on some coupling constants, which correspond to the connection weights between neurons. Tracing out UV degrees of freedom then corresponds to marginalizing over some subset of neurons $\mathbf{x}\backslash\mathbf{x}' \subset \mathbf{x}$ (where $\mathbf{x}\backslash\mathbf{x}'$ is the complement of $\mathbf{x}' \subset \mathbf{x}$), subject to the constraint that the partition function is preserved, i.e.,

$$p(\mathbf{x}') = \text{tr}_{\mathbf{x}\backslash\mathbf{x}'} \, p(\mathbf{x}) = Z^{-1} \, \text{tr}_{\mathbf{x}\backslash\mathbf{x}'} \, e^{-H(\mathbf{x})} =: Z^{-1}e^{-H'(\mathbf{x}')} \,, \tag{1}$$

where $H'(\mathbf{x}')$ is the coarse-grained Hamiltonian defined in terms of the remaining neurons, which play the role of the IR degrees of freedom. However, this is merely a structural analogy that holds for *any* (Bayesian) hierarchical model (including deep or stacked RBMs, as one can see by iteratively applying (1) at subsequent layers). Contrary to some suggestions in the literature, this formal analogy with RG does not suffice to explain how deep neural networks learn. During the learning process, the couplings in the Hamiltonian are dynamically updated, whereas these are fixed under RG via the constraint (1). Nonetheless, the analogy may be helpful for understanding the behavior and properties of these networks. In particular, one interesting direction is to examine the flow of cumulants in successive layers, which can

---

[1]Here, we mean "abstract" in the usual sense for such hierarchical systems; e.g., relative to a stream of pixel values, higher-level features and objects (such as shapes or faces) are considered more abstract. An outstanding question in machine learning related to this work is to quantify abstraction in deep networks; see for example [11, 12].

induce higher-order interactions that encode the correlations between marginalized degrees of freedom [8–10].

It is therefore interesting to examine this structural analogy in more detail, in particular with an eye towards understanding how "information" is processed at subsequent RG steps/network layers. The fact that information is lost under RG is well-known: intuitively, RG may be thought of as a coarse-graining procedure, in which fine-grained information gets traced out or marginalized over at low energies. In relativistic quantum field theory, this corresponds to the fact that high-energy degrees of freedom decouple, and the total number of degrees of freedom decreases as one moves into the IR. This has been quantified by Zamolodchikov's famous $c$-theorem and various extensions [14–21], which define a quantity which decreases monotonically under RG. The $c$-function can be straightforwardly related to entanglement entropy in certain cases (see for example [17,18]). Alternatively, it may be related to relative entropy [22], which we define momentarily and which plays an important role in our analysis.

For neural networks however, there is so far no general quantification of hierarchical abstraction in terms of information-theoretic language. An ultimate goal would be to not only quantify the flow of information through subsequent layers, but also to find measures to qualify the learning process. While some progress in this direction has been achieved (see for example [3,10–12,23–29] and references therein), this goal has not yet been universally achieved. We expect that further insight from physics may be of use in this context.

In this paper, we pursue a more modest goal of quantifying the information flow in these two systems, using the relative entropy or Kullback-Leibler (KL) divergence,

$$D(p||q) = \mathrm{tr}_{\mathbf{x}}\, p(\mathbf{x}) \ln \frac{p(\mathbf{x})}{q(\mathbf{x})} \,, \tag{2}$$

which quantifies the extent to which the target distribution $q$ differs from some reference distribution $p$. The KL divergence is non-negative[2], and equals zero only when the two distributions are identical. Note that insofar as the KL divergence is asymmetric and fails to satisfy the triangle inequality, it is not a metric distance[3]; nonetheless, it provides a measure of the difference in information content of $q$ relative to $p$. In the present context for example, $p$ corresponds to the UV theory at equilibrium, or to the first (input) layer of a deep neural network; $q$ then respectively represents the theory at some point along the RG flow to the IR, or the distribution of neurons on an arbitrary hidden layer[4]. We then wish to understand how the relative entropy behaves as a function of depth – both along the RG flow, and when moving to deeper and deeper layers of the network – as a means of quantifying information in such systems.

Specifically, we compute the KL divergence explicitly for the 1- and 2-dimensional Ising models under decimation RG on the one hand, and a simple feedforward random network[5] on the other, and examine the similarities between the two. The results in both cases are qualitatively identical: the KL divergence rapidly and monotonically increases before asymptoting to some value that depends on either the coupling constants (in the Ising model) or the parameters characterizing the weights and biases (in the neural network). For the Ising models, the only work of which we are aware that studies a similar (though not identical) KL divergence under decimation RG is Fowler's 2020 thesis [31]; we will comment on the differences

---

[2]Proofs of non-negativity implicitly assume that both distributions are normalized with respect to the same measure. As we will discuss in sections 2 and 3, the changing dimensionality of the Hilbert space must be taken into account in order to avoid a non-monotonic, potentially negative result.

[3]It is, however, intimately related to the Fisher information metric; see for example [30] for a recent exploration in the context of field theory.

[4]We will give an argument in section 2 below as to why it is incorrect to invert the identifications such that $p$ instead represents the IR.

[5]This class of deep neural network will be reviewed in section 3.

between our analyses in section 2. For the neural networks, as far as we are aware, this is the first such explicit computation to have appeared in the literature, though the use of various entropic quantities in machine learning has a deep and diverse history. For example, the entropy of a given layer, as well as the closely-related mutual information, have recently been evaluated via replica methods in [32]. Entropy has also been proposed as a training mechanism in the context of adversarial learning in [33]; see also [34,35]. Additionally, estimates of $f$-divergences (of which the KL divergence is perhaps the canonical example) are relevant for generative adversarial networks, see e.g., [36]. Mutual information in particular – which may be defined as the KL divergence of $p(x)p(y)$ relative to $p(x, y)$ – has a wide utility ranging from the information bottleneck [37] to various maximization (training) methods [38–42]. For a non-exhaustive sampling of other works on estimating or bounding entropic quantities in deep learning, see [43–47].

Interestingly, while we shall give some further arguments below as to why the monotonicity may be expected[6], we find the KL divergence to be insensitive to the phase behavior of the system. The 2d Ising model, for example, exhibits a continuous phase transition from ordered to disordered at some finite value of the couplings, whereas the behavior of the KL divergence shows no apparent change as we smoothly dial the couplings through this point. Similarly, as we will review in section 3, recent work by [23, 24] showed that the feedforward neural networks under study also exhibit such a phase transition, which appears to control the depth to which a given network can be trained (see also [25–27], for extensions of this work to more complicated architectures, or [28, 48] for reviews). The basic intuition is that the critical point is characterized by a divergent correlation length, which allows information about the input data to propagate all the way through a network of theoretically arbitrary depth. However, quantifying precisely what this means in information-theoretic language is challenging, and indeed one original motivation for this work was to determine whether the Kullback-Leibler divergence could provide a complementary or alternative characterization of information propagation in this context.

Before entering into the heart of our analysis, let us mention that the relation between relative entropy and RG flow in the context of quantum field theory has been explored in a different setting in [49]. There, relative entropy was used to define a proximity notion between probability distributions associated to different QFTs. Its second derivative with respect to the RG scale was related to the Zamolodchikov metric.

This paper is organized into two main sections: in section 2, we compute the relative entropy (KL divergence) for both the 1d and 2d classical Ising models under decimation RG. Specifically, we fix the reference distribution $p$ to be the initial system, and take the target distribution $q$ to be the system under sequential RG steps. In section 3, we first introduce the feedforward, random neural networks of the same type studied in [23, 24], and briefly summarize the notion of criticality in these networks for the sake of completeness. We then compute the KL divergence as a function of depth by fixing the reference distribution to be that of the neurons comprising the input layer, and move the target distribution through all subsequent (hidden) layers. We conclude with some discussion in section 4. For the sake of completeness, we have included two short appendices: a review of the real-space decimation procedure for the 1d Ising model in appendix A, and some details about the evaluation of the KL divergence in our deep neural networks via Monte Carlo integration in appendix B.

---

[6]We note that the monotonicity of relative entropy has been proven for perturbed 2d CFTs in [22].

## 2 Relative entropy of spin models under decimation

In this section, we will examine the relative entropy or Kullback-Leibler (KL) divergence between different points along the RG flow of the 1d and 2d Ising models. Specifically, we will perform the RG flow in real-space via the standard decimation procedure, and track the KL divergence of the coarse-grained or infra-red (IR) distribution of spin states at successive steps relative to the initial fine-grained or ultra-violet (UV) distribution.

The decimation RG procedure applied to the Ising models has been well studied since the introduction of the "block spin" concept by Kadanoff in 1966 [50]. There are many excellent articles and reviews we could cite for this topic, but we will limit ourselves to those we actively referenced in the course of this work. These are Wilson's seminal work on renormalization and critical phenomena [51], the early pedagogical review of renormalization by Maris and Kadanoff [52], the statistical mechanics text by Pathria [53], and Vvedensky's course notes[7] [54]. In addition, we will of course refer to Onsager's exact solution of the 2d Ising model [55].

We begin with a general discussion of calculating this relative entropy for a general spin system. Let the spins be labeled by some index $i$ as $\sigma_i = \pm 1$ and let $H(\sigma_i)$ be the Hamiltonian of the system. The partition function is $Z = \sum e^{-H(\sigma_i)}$, where the sum is over the entire Hilbert space $\mathcal{H}$ of spin states of the system (i.e., $\sum_{\{\sigma\}} \equiv \prod_i \sum_{\sigma_i = \pm 1}$), and we have absorbed the inverse temperature $\beta$ into the couplings[8]. The state of the system at any given time follows a Boltzmann distribution,

$$p(\sigma_i) = \frac{1}{Z} e^{-H(\sigma_i)}. \tag{3}$$

As a probability distribution function, $p$ is an element of all positive semi-definite normalized real functions on $\mathcal{H}$.

Decimation is then defined by some map $\sigma'_j(\sigma_i)$ from $\mathcal{H}$ to a subset $\mathcal{H}' \subset \mathcal{H}$. Formally, $\mathcal{H}'$ should be strictly smaller than $\mathcal{H}$ in the sense that the difference is nonempty $\mathcal{H} \backslash \mathcal{H}' \neq \emptyset$. In Wilsonian language, this corresponds to reducing the energy scale at which one probes the system, such that the fine-grained degrees of freedom – in this case, the elements in $\mathcal{H} \backslash \mathcal{H}'$ of $H(\sigma_i)$ – have been marginalized over in the effective Hamiltonian $H'(\sigma'_j)$ on $\mathcal{H}'$. A key fact of the renormalization group is that it preserves the partition function, i.e.,

$$Z = \sum_{\sigma_i \in \mathcal{H}} e^{-H(\sigma_i)} = \sum_{\sigma'_j \in \mathcal{H}'} \sum_{\sigma_i \in \mathcal{H} \backslash \mathcal{H}'} e^{-H(\sigma_i)} = \sum_{\sigma'_j \in \mathcal{H}'} e^{-H'(\sigma'_i)} = Z', \tag{4}$$

where the identification in the penultimate step follows from the fact that the IR distribution $p'(\sigma')$ is obtained by marginalizing or tracing over $\sigma_i \in \mathcal{H} \backslash \mathcal{H}'$ in the UV distribution $p(\sigma_i)$:

$$p'(\sigma'_j) = \sum_{\sigma_i \in \mathcal{H} \backslash \mathcal{H}'} p(\sigma_i) = \frac{1}{Z} \sum_{\sigma_i \in \mathcal{H} \backslash \mathcal{H}'} e^{-H(\sigma_i)} =: \frac{1}{Z} e^{-H'(\sigma'_j)}. \tag{5}$$

A crucial consequence of this structure, underlying both our analysis below and our ability to do physics in general, is that the expectation value of an IR observable $\mathcal{O}'$ with respect to $p'$ is the same as that with respect to $p$:

$$\begin{aligned} \langle \mathcal{O}' \rangle_p &= \frac{1}{Z} \sum_{\sigma_i \in \mathcal{H}} e^{-H(\sigma_i)} \mathcal{O}'(\sigma'_j(\sigma_i)) = \frac{1}{Z} \sum_{\sigma'_j \in \mathcal{H}'} \mathcal{O}'(\sigma'_j) \sum_{\sigma_i \in \mathcal{H} \backslash \mathcal{H}'} e^{-H(\sigma_i)} \\ &= \frac{1}{Z} \sum_{\sigma'_j \in \mathcal{H}'} \mathcal{O}'(\sigma'_j) e^{-H'(\sigma'_j)} = \langle \mathcal{O}' \rangle_{p'}. \end{aligned} \tag{6}$$

---

[7]We are grateful to Dimitri Vvedensky for his correspondence and course notes.

[8]In our numerical experiments below, we will effectively set this to 1 along with Boltzmann's constant.

Note that this does not work in reverse: it is incorrect to compute $\langle \mathcal{O} \rangle_{p'}$ because a UV observable $\mathcal{O}$ will generically depend on fine-grained information about which the IR distribution $p'$ is ignorant.

We can then formally define the entropy of the distribution after decimation relative to the distribution before decimation as a slight modification of (2):

$$S(p||p') := \sum_{\sigma_i \in \mathcal{H}} p(\sigma_i) \ln\left( \frac{p(\sigma_i)}{p'\big(\sigma_j'(\sigma_i)\big)} \right) + \ln|\mathcal{H}\backslash\mathcal{H}'|, \tag{7}$$

where $|\mathcal{H}\backslash\mathcal{H}'|$ is the dimension of the complement of $\mathcal{H}'$ in $\mathcal{H}$. The reason why this extra term must be added is due to the fact that, while $p$ is properly normalized with respect to $\mathcal{H}$, the distribution $p'(\sigma_j')$ is properly normalized only on $\mathcal{H}'$. If we think of the spins $\sigma_j'$ as being a map from $\mathcal{H}$ to $\mathcal{H}'$, then this map is many-to-one and, therefore, thought of as a distribution on $\mathcal{H}$, the integral of $p'$ is not unity, but rather the dimension of the complement of $\mathcal{H}'$:

$$\int_{\mathcal{H}} p'\big(\sigma_j'(\sigma_i)\big) = \int_{\mathcal{H}\backslash\mathcal{H}'} \int_{\mathcal{H}'} p'(\sigma_j') = \int_{\mathcal{H}\backslash\mathcal{H}'} 1 = |\mathcal{H}\backslash\mathcal{H}'|. \tag{8}$$

In other words, $p'\big(\sigma_j'(\sigma_i)\big)$ must be divided by $|\mathcal{H}\backslash\mathcal{H}'|$ to be a normalized distribution over all of $\mathcal{H}$, which leads to the extra term in the relative entropy (7). A similar normalization issue will arise when we consider the relative entropy in deep neural networks in section 3; see the discussion around (91) therein.

Due to the fact that the decimation map preserves the partition function, the expression for the relative entropy simplifies to

$$\begin{aligned} S(p||p') &= \frac{1}{Z} \sum_{\sigma_i \in \mathcal{H}} e^{-H(\sigma_i)} \big[ H'\big(\sigma_j'(\sigma_i)\big) - H(\sigma_i) \big] + \ln|\mathcal{H}\backslash\mathcal{H}'| \\ &= \langle H' \rangle_{p'} - \langle H \rangle_p + \ln|\mathcal{H}\backslash\mathcal{H}'|, \end{aligned} \tag{9}$$

where on the second line, we have used the observation (6) that $\langle H' \rangle_p = \langle H' \rangle_{p'}$.

## 2.1  1d classical Ising model

Let us now apply the formalism above to the 1d classical Ising model with $N$ spins:

$$H = -\sum_{i=1}^{N} \big( K_0 + K \sigma_i \sigma_{i+1} \big), \tag{10}$$

where we impose the periodic boundary condition $\sigma_{N+1} = \sigma_1$. As usual, we expect the choice of boundary condition to be irrelevant in the thermodynamic limit $N \to \infty$.

To minimize technical complications, we shall perform the simplest decimation procedure by which we sum over the spins at the even lattice sites:

$$\sigma_j' = \sigma_{2j-1}, \tag{11}$$

i.e., $\mathcal{H}' = $ odd spins and $\mathcal{H}\backslash\mathcal{H}' = $ even spins, with dimensions $|\mathcal{H}'| = |\mathcal{H}\backslash\mathcal{H}'| = 2^{\frac{N}{2}}$. After some standard algebra (see appendix A), one arrives at the result for the new Hamiltonian

$$H' = -\sum_{j=1}^{N/2} \big( K_0' + K' \sigma_j \sigma_{j+1} \big), \tag{12}$$

where

$$K' = \frac{1}{2}\ln\cosh(2K), \tag{13a}$$

$$K_0' = \ln 2 + 2K_0 + \frac{1}{2}\ln\cosh(2K). \tag{13b}$$

Similarly, let $K^{(n)}$ and $K_0^{(n)}$ denote the result of iterating this recursion relation $n$ times:

$$K^{(n)} = \frac{1}{2}\ln\cosh(2K^{(n-1)}) = \frac{1}{2}\ln\cosh\big(\ln\cosh(2K^{(n-2)})\big) = \cdots, \tag{14a}$$

$$K_0^{(n)} = (2^n - 1)\ln 2 + 2^n K_0 + 2^n \sum_{m=1}^{n}\frac{K^{(m)}}{2^m}, \tag{14b}$$

where $2K^{(n)}$ can be written as $m$ nested $\ln\cosh$'s acting on $2K^{(n-m)}$ for $m = 1,\ldots,n$.

Next, we must compute the expectation values of the Hamiltonians with respect to their respective Boltzmann distributions. This is straightforward to obtain from the reduced free energy per site $f := \frac{\beta}{N}F = -N^{-1}\ln Z$, which one can find in any decent statistical mechanics textbook:

$$f(K_0, K) = -\ln 2 - K_0 - \ln\cosh K. \tag{15}$$

As discussed in the introduction, the partition function $Z$ must be preserved under the decimation procedure. While this is not obvious at the level of the iterated recursion relations (14), we have proven it explicitly in appendix A.1.

The expectation value of $H$ with respect to $p$ is then

$$\begin{aligned}\langle H\rangle_p &= -NK_0 - K\left\langle\sum_{i=1}^{N}\sigma_i\sigma_{i+1}\right\rangle_p = -NK_0 - K\partial_K\ln Z \\ &= -NK_0 + NK\partial_K f = -N(K_0 + K\tanh K),\end{aligned} \tag{16}$$

where in the first term we have used the fact that the distribution is properly normalized, i.e., $\langle 1\rangle = \frac{1}{Z}\mathrm{tr}_\sigma e^{-H} = 1$. Similarly,

$$\langle H'\rangle_{p'} = -\frac{N}{2}\big(K_0' + K'\tanh K'\big). \tag{17}$$

For convenience, let us define the *normalized* relative entropy $s(p\|p')$ to be the relative entropy divided by the total entropy of the initial (reference/UV) system in the infinite-temperature (i.e., $K \to 0$) limit, which in the present case is simply

$$S_0 = N\ln 2. \tag{18}$$

Thus, after a single decimation step, we have

$$s(p\|p') := \frac{S(p\|p')}{S_0} = \frac{1}{\ln 2}\left(K_0 + K\tanh(K) - \frac{1}{2}K_0' - \frac{1}{2}K'\tanh(K')\right) + \frac{1}{2}. \tag{19}$$

If we then iterate this procedure $n$ times, the dimensions of the Hilbert spaces become

$$|\mathcal{H}^{(n)}| = 2^{\frac{N}{2^n}}, \qquad\qquad |\mathcal{H}\backslash\mathcal{H}^{(n)}| = 2^{N(1-\frac{1}{2^n})}, \tag{20}$$

and the normalized relative entropy after $n$ RG steps is

$$s^{(n)} := s\big(p\|p^{(n)}\big) = \frac{1}{\ln 2}\left(K_0 + K\tanh(K) - \frac{1}{2^n}K_0^{(n)} - \frac{1}{2^n}K^{(n)}\tanh\big(K^{(n)}\big)\right) + 1 - \frac{1}{2^n}, \tag{21}$$

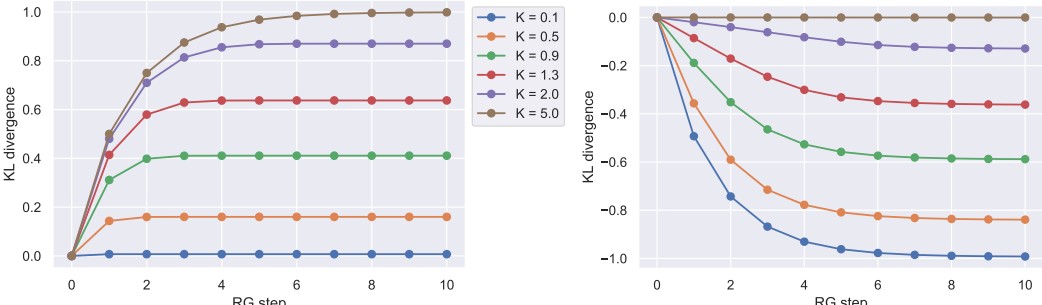

Figure 1: Relative entropy (Kullback-Leibler divergence) of the 1d classical Ising model as a function of real-space decimation step for various values of the nearest-neighbor coupling $K$. The left plot shows the correctly normalized entropy, while the right shows the same data without the normalization factor.

where $K_0^{(n)}$ and $K^{(n)}$ are the couplings after $n$ decimation steps, given in (14). Note that $K_0$ drops out of the final expression; this is to be expected, since $K_0$ simply represents an arbitrary choice for the zero-point of the free energy. The plot of this normalized relative entropy as a function of the decimation step for various values of the nearest-neighbor interaction coupling $K$ is shown in the left panel of fig. 1.

To provide some physical insight into this result, recall that the extra $\ln |\mathcal{H} \backslash \mathcal{H}'|$ term in the definition of the relative entropy (9) – which gives rise to the $1 - \frac{1}{2^n}$ in (21) – is purely due to the fact that the decimation procedure reduces the dimensionality of the system by half. Had we not included this term, we would instead obtain the result shown in the right panel of fig. 1. The bottom curve, corresponding to weak coupling $K \ll 1$, is then quite easy to interpret: at weak coupling, the spins hardly communicate with one another, and therefore the effect of decimation is to simply halve the number of degrees of freedom – i.e., halve the information content – at each step. Obviously, we cannot lose any more information than was present in the initial configuration to begin with, which in the small-$K$ limit is simply the total entropy of $N$ independent spins, $S_0 = N \ln 2$. This is why the bottom curve on the right plot in fig. 1 is bounded below by $-1$, as well as our reason for defining the normalized relative entropy above.

Now, as we increase the coupling, the spins become (classically) correlated, and hence the total information content of the system consists in part of information stored in these correlations. The stronger the coupling, the more information can be stored in long-range correlations, and hence the more robust it will be against decimation. This is why the top curve on the right plot in fig. 1 is approximately 0: the monotonic reduction in the number of degrees of freedom does not noticeably alter the information content, since as $K \to \infty$ the correlations become so strong that the system behaves like a single collective mode.

We therefore observe a clear connection between the relative entropy without the extra normalization factor $\ln |\mathcal{H} \backslash \mathcal{H}'|$ and the celebrated $c$-theorem, in which the eponymous $c$-function measures the number of degrees of freedom [14–19, 21]. The fact that it decreases monotonically under RG from the UV to the IR precisely and rigorously demonstrates the intuitive idea that the number of degrees of freedom decreases as we flow to low energies[9]. Indeed, the monotonic increase in the relative entropy observed in the left plot of fig. 1 is obtained only when accounting for this reduction in dimensionality.

---

[9]Here, we refer to the $c$-theorem in the broader sense that there exists a function measuring the number of degrees of freedom that decreases along the RG flow. We are not claiming a direct relationship with the Zamolodchikov $c$-function in perturbed CFTs.

In fact, it is easy to prove that the normalized relative entropy must be monotonically increasing. First, observe that repeated application of (13) yields

$$K_0^{(n)} = 2^n K_0 + 2^n \sum_{m=1}^n \frac{K^{(m)}}{2^m} + (2^n - 1) \ln 2 \,. \tag{22}$$

Substituting this into (21), we have an expression for the normalized relative entropy in terms of the coupling $K$ at every decimation step:

$$s^{(n)} = \frac{1}{\ln 2}\left( K \tanh(K) - \frac{1}{2^n} K^{(n)} \tanh\!\left(K^{(n)}\right) - \sum_{m=1}^n \frac{K^{(m)}}{2^m} \right). \tag{23}$$

Hence, the difference between successive decimation steps satisfies the relation

$$\left(2^{n+1} \ln 2\right)\!\left(s^{(n+1)} - s^{(n)}\right) = 2K^{(n)} \tanh\!\left(K^{(n)}\right) - K^{(n+1)}\!\left[1 + \tanh\!\left(K^{(n+1)}\right)\right]. \tag{24}$$

Since $\tanh x < 1$ for all finite $x$,

$$
\begin{aligned}
\left(2^{n+1} \ln 2\right)\!\left(s^{(n+1)} - s^{(n)}\right) &> 2K^{(n)} \tanh\!\left(K^{(n)}\right) - 2K^{(n+1)} \\
&= 2K^{(n)} \tanh\!\left(K^{(n)}\right) - \ln \cosh\!\left(2K^{(n)}\right).
\end{aligned}
\tag{25}
$$

The last step is to observe that the function on the far right-hand side, $2x \tanh(x) - \ln \cosh(2x)$, is $\geq 0$ for all $x \geq 0$ since its value at $x = 0$ is 0 and its derivative is $2x \operatorname{sech}^2(x)$, which is $\geq 0$ for all $x \geq 0$. Therefore, we have proven that the normalized relative entropy monotonically increases under decimation RG for non-vanishing values of the coupling $K$, i.e.,

$$s^{(n+1)} - s^{(n)} > 0 \,. \tag{26}$$

This analysis is in line with the result of [22] based on modular Hamiltonians in perturbed 2d CFTs.

**Weakly-coupled limit**

We can go slightly further in our discussion of weak- vs. strong-coupling above and derive some approximate expressions for the relative entropy in these limits. When $K \ll 1$, the recursion relations (13) read

$$K' = K^2 + O(K^4), \tag{27a}$$
$$K_0' = 2K_0 + \ln 2 + K^2 + O(K^4). \tag{27b}$$

Repeated iteration gives

$$K^{(n)} = K^{2n} + O(K^{2(n+1)}), \tag{28a}$$
$$K_0^{(n)} = 2^n K_0 + (2^n - 1) \ln 2 + 2^{n-1} K^2 + O(K^4). \tag{28b}$$

Plugging these into the normalized relative entropy, we find

$$s^{(n)} = \frac{K^2}{2 \ln 2} + O(K^4), \qquad n \geq 1 \,, \tag{29}$$

where $s^{(0)} = 0$. Thus, after the first decimation step, the normalized relative entropy remains approximately constant, and simply scales quadratically with the strength of the nearest-neighbor interaction.

**Strongly-coupled limit**

In the strongly-coupled limit, when $K \gg 1$, the recursion relations (13) instead read

$$K' = K - \frac{1}{2}\ln 2 + O(e^{-4K}), \tag{30a}$$

$$K_0' = 2K_0 + K + \frac{1}{2}\ln 2 + O(e^{-4K}). \tag{30b}$$

Repeated iteration gives

$$K^{(n)} = K - \frac{n}{2}\ln 2 + O(e^{-4K}), \tag{31a}$$

$$K_0^{(n)} = 2^n K_0 + (2^n - 1)K + \frac{n}{2}\ln 2 + O(e^{-4K}), \tag{31b}$$

whence the normalized relative entropy becomes

$$s^{(n)} = 1 - \frac{1}{2^n} - (n\ln 2)e^{-2K} + O(e^{-4K}). \tag{32}$$

However, this is an asymptotic expansion which predicts its own breakdown at around[10]

$$n_* \approx \frac{2K}{\ln 2}, \tag{33}$$

at which point $K^{(n)} \approx 0$, such that the large-$K$ approximation is no longer possible.

**Asymptote of the relative entropy**

Let us now ask about the asymptotic value of $s^{(\infty)}$ itself. We know that it must behave like (29) for small $K$, and that it must go to unity as $K \to \infty$ due to the normalization, cf. (18). However, while we are unable to evaluate the asymptote analytically in the large-$K$ limit, we can obtain a surprisingly good fit with the following ansatz:

$$s_{\text{fit}}^{(\infty)} = \frac{\frac{K^2}{2\ln 2}f(K)}{1 + \frac{K^2}{2\ln 2}f(K)}, \tag{34}$$

where $f(K)$ is function to be determined. In the limit $K \to \infty$, the only restriction on $f(K)$ is that it must not go to zero faster than $\frac{1}{K^2}$, so that

$$s_{\text{fit}}^{(\infty)} \xrightarrow{K\to\infty} 1. \tag{35}$$

Meanwhile, to have the correct small-$K$ limit (29), the behavior of $f(K)$ as $K \to 0$ must be

$$f(K) \xrightarrow{K\to 0} 1. \tag{36}$$

We then compare various choices of $f(K)$ satisfying these two limits to the data obtained above. By inspection of fig. 1, $n = 10$ is a sufficient approximation to $n = \infty$ for the present purpose. After some experimentation, we obtain the following function fit:

$$s_{\text{fit}}^{(\infty)} = \frac{\frac{K^2}{2\ln 2}\cosh\frac{K}{2\ln 2}}{1 + \frac{K^2}{2\ln 2}\cosh\frac{K}{2\ln 2}}. \tag{37}$$

This is plotted in fig. 2, which shows remarkably good agreement with the numerical results.

---

[10]Strictly speaking, since $n_* \in \mathbb{N}$, one should apply the floor function to the right-hand side.

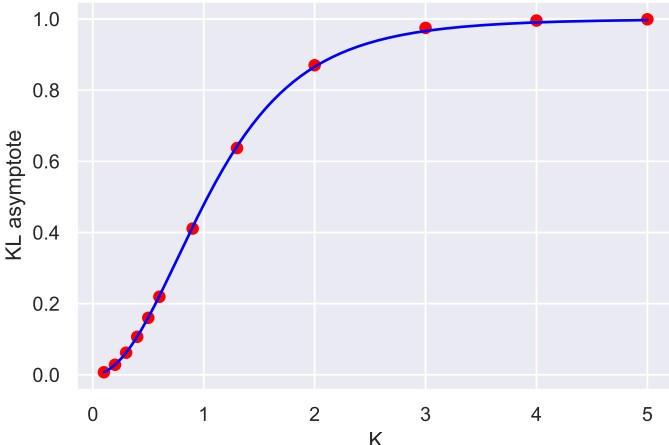

Figure 2: The asymptote of the normalized relative entropy of the 1d classical Ising model as a function of the nearest-neighbor interaction coupling constant $K$ (red dots). The solid line is the fit function (37).

**Comparison with previous work**

After completing this work, we became aware of the thesis by Fowler [31], in which a quantity called the "KL density" (KL divergence per spin) is computed for the 1d and 2d Ising models under decimation RG. In that case however, what the author refers to as the KL divergence is in fact the mutual information (MI) between the joint distribution of spins for the initial system and the product distribution of spins at subsequent decimation steps. Both quantities – the KL divergence we compute here, and the MI in figs. 8.1 and 8.2 of [31] – increase monotonically to an asymptotic value which itself increases with increasing coupling constant $K$. In fig. 1 (left plot), we have normalized by the entropy of $N$ spins, cf. (18), so that the $K \to \infty$ asymptote approaches unity, in contrast to Fowler's $\ln 2 \approx 0.69$.

An important difference between these quantities is that the relative entropy is not only monotonically increasing as a function of decimation step but, at any fixed decimation step, is also monotonically increasing as a function of $K$. Therefore, it displays no qualitative difference in behavior as we move towards the unstable fixed point at $K \to \infty$. In contrast, the mutual information in [31] takes more and more decimation steps for the curves to rise as we increase $K$; that is, stronger coupling delays the increase in the mutual information.

Conceptually, another difference is the relation to the notion of information loss under RG that we discussed in the introduction. While the KL divergence provides a direct measurement of the total information content as a function of RG step, the MI probes the information stored in correlations between the product distributions. *A priori*, this may increase, decrease, or remain unchanged depending on the system at hand, so the relation between RG and the monotonicity we observed for the KL divergence above does not necessarily hold for a generic hierarchical system. In fact, for the neural networks we study in section 3, this mutual information is identically zero for all layers, since the distribution on each layer factorizes. Thus the KL divergence (2) is a more useful quantity for our purposes, since it allows us to compare the similarities between these different systems.

## 2.2 2d classical Ising model

While the one-dimensional Ising model provides a clean proof of concept – as well as the most obvious parallel to the feedforward networks we shall consider in section 3 – the lack of a non-

trivial critical point means that we cannot investigate whether the relative entropy is sensitive to the phase structure of the theory. We therefore move to the two-dimensional classical Ising model, which exhibits a well-known phase transition at finite temperature.

For simplicity, we shall consider the isotropic case on a square lattice of $N$ spins, so that the Hamiltonian is

$$H = -NK_0 - K \sum_{\text{n.n.}} \sigma_i \sigma_j \, , \tag{38}$$

where the sum runs over nearest-neighbor pairs, and we impose the usual periodic boundary conditions, which are again inconsequential in the thermodynamic limit $N \to \infty$.

We consider the standard "checker board" decimation procedure whereby we marginalize over the spins whose lattice coordinates sum to an odd number, i.e., every-other spin in both the horizontal and vertical directions. Unfortunately, the Hamiltonian is not closed under this decimation procedure (that is, more complicated interaction terms are generated at each decimation step)[11]. For example, at the first decimation step, a next-to-nearest neighbor interaction term as well as a four-point "square plaquette" interaction term are generated; the associated couplings are denoted $L'$ and $M'$ in Pathria's text, respectively [53], so that after one decimation step, the Hamiltonian becomes

$$H' = -\frac{N}{2} K_0' - K' \sum_{\text{n.n.}} \sigma_i' \sigma_j' - L' \sum_{\text{n.n.n.}} \sigma_i' \sigma_j' - M' \sum_{\text{sq.}} \sigma_i' \sigma_j' \sigma_k' \sigma_\ell' \, , \tag{39}$$

where n.n.n. stands for next-to-nearest-neighbors and sq. stands for square plaquette. The new parameters are related to the initial (UV) coupling $K$ by

$$K_0' = 2K_0 + \ln 2 + \frac{1}{2} \ln \cosh(2K) + \frac{1}{8} \ln \cosh(4K) \, , \tag{40a}$$

$$K' = \frac{1}{4} \ln \cosh(4K) \, , \tag{40b}$$

$$L' = \frac{1}{8} \ln \cosh(4K) \, , \tag{40c}$$

$$M' = \frac{1}{8} \ln \cosh(4K) - \frac{1}{2} \ln \cosh(2K) \, . \tag{40d}$$

Let us make a few comments at this point. First, we cannot solve this set of coupled equations analytically. Additionally, we observe that there is no finite solution for the critical value $K_c$, since the only solutions to $K' = K$ are $K = 0$ and $K = \infty$. Therefore, even trying to solve these equations numerically would be ineffectual. Second, since longer-range interactions represented by the $L'$ and $M'$ terms are generated after decimation, one should have introduced such terms in the initial Hamiltonian (38) for consistency. However, more and more complicated interactions will be generated under each subsequent decimation step, and therefore no finite truncation of these interactions is strictly consistent. Nevertheless, we expect that there should be a regime in which higher-order interactions are relatively weak, and the dynamics are primarily governed by the nearest-neighbor term. Furthermore, this hierarchy of interaction scales should be preserved under RG[12]. With this in mind, several approximations have been proposed to obtain useful recursion relations from (40).

---

[11]That real-space RG is generally problematic has been known since the work of Griffiths and Pearce [56,57]; see also [58,59]. Physically, the intuition is that in momentum space, the RG should remove only those modes which lie above some UV cut-off, corresponding to the inverse lattice spacing. If however we marginalize over, say, every-other spin, we have removed not only the nearest-neighbor correlations at the cut-off scale, but also some of the long-range correlation between distant marginalized spins.

[12]If this were not the case, then there would be no way to make sense of the pure 2d classical Ising model at all, since higher-order terms would become increasingly important at low energies.

The approach originally proposed by Wilson in 1975 [51] is to ignore all interaction terms except for $K$ and $L$, introduce an $L$ interaction term in the Hamiltonian $H$ to begin with, and assume that $K, L \ll 1$ so that the recursion relations may be expanded. Another approach proposed by Maris and Kadanoff [52] is to ignore $M'$ and to define an "effective" nearest-neighbor coupling after decimation to be the sum of $K'$ and $L'$, since having a next-to-nearest-neighbor interaction is somewhat analogous to having a slightly stronger nearest-neighbor interaction. There is a slightly more detailed energetic argument behind this idea in the original paper by Maris and Kadanoff, to which we refer the interested reader. In this approach, the recursion relations read

$$K_0' = 2K_0 + \ln 2 + \frac{1}{2}\ln\cosh(2K) + \frac{1}{8}\ln\cosh(4K),\tag{41a}$$

$$K' = \frac{3}{8}\ln\cosh(4K),\tag{41b}$$

where $K'$ here is the effective $K'$, being the sum of $K'$ and $L'$ in (40).

Both of these methods modify the recursion relation (40b) slightly so as to produce a fixed point at some finite value of $K$, called the critical coupling $K_c$. Without modification, the only fixed points (i.e., solutions to $K' = K$) are $K = 0$ and $K = \infty$, with $K = 0$ being stable and $K = \infty$ being unstable. In the Maris-Kadanoff approach there is a third fixed point at

$$K_c^{\text{Maris-Kadanoff}} \approx 0.506981,\tag{42}$$

which is now the unstable fixed point, while $K = 0$ and $K = \infty$ are now both stable.

In the approach of Wilson, the story is slightly more complicated. We will summarize the main result here and refer the reader to [51,53] for the details. There is a fixed point in the $(K, L)$-plane at $K^* = \frac{1}{3}$ and $L^* = \frac{1}{9}$ and one critical line of points that flow to this fixed point under renormalization. This line is extrapolated linearly to the $K$-axis, where $L = 0$, and the intersection is the critical value $K_c$, which gives

$$K_c^{\text{Wilson}} \approx 0.397904.\tag{43}$$

Meanwhile, the exact solution for the critical coupling due to Onsager is [55]

$$K_c = \frac{1}{2}\text{arcsinh}(1) \approx 0.440687.\tag{44}$$

To somewhat tie these various approaches together, one could consider a one-parameter extension of the Maris-Kadanoff approach in which we replace (41) with

$$K_0' = 2K_0 + \ln 2 + \frac{1}{2}\ln\cosh(2K) + \frac{1}{8}\ln\cosh(4K),\tag{45a}$$

$$K' = \frac{\alpha}{4}\ln\cosh(4K),\tag{45b}$$

where $\alpha$ is a positive real number. The original recursion relation (40) corresponds to the value $\alpha = 1$, while completely ignoring $L'$ and $M'$. Again, this has no non-trivial fixed points besides $K = 0$ and $K = \infty$. The modification due to Maris and Kadanoff in (41) corresponds to $\alpha = \frac{3}{2}$, which gives a non-trivial fixed point at (42). The value $\alpha^*$ of $\alpha$ that would produce the exact critical coupling (44) is slightly greater than this:

$$\alpha^* = 1.604521.\tag{46}$$

All of these approaches suffer at large coupling. This is a fundamental drawback of real space decimation RG when applied to the 2d Ising model; see footnote 11, which offers some intuition for the fact that problems are more pronounced when $K$ is large.

However the model is exactly solvable by the methods first developed by Onsager [55], so we know $K_c$ exactly (44), and we know that this fixed point is repulsive on both sides: if $K < K_c$, then the system flows towards the $K = 0$ free fixed point and if $K > K_c$, then the system flows towards the $K = \infty$ strongly-coupled fixed point. Therefore, in the region $0 \leq K \leq K_c$, it is justified to ignore all the complicated additional interaction terms that are generated at each decimation step. Above $K_c$ however, $K$ tends to grow after each decimation step, which means that the additional interactions (e.g., the $M$-type interaction), will also tend to grow in magnitude. Ignoring these interactions will thus result in an increasingly bad approximation at larger and larger $K$. This will become patently clear when we plot the normalized relative entropy for the 2d Ising model at strong coupling below (see fig. 4): if we push $K$ above the critical value, the *normalized* relative entropy begins to decrease, and eventually becomes negative as $K \to 1$, signalling an obvious breakdown of the approximation.

Now, to calculate the relative entropy, we need the expectation value of the Hamiltonian. As before, we start with the reduced free energy per site, which for the isotropic model with no external magnetic field, reads

$$f = \frac{1}{2}\ln 2 + \frac{1}{2\pi}\int_0^\pi d\phi \, \ln\left(\cosh^2(2K) + \sqrt{1 + \sinh^4(2K) - 2\sinh^2(2K)\cos(2\phi)}\right). \quad (47)$$

Then, the expectation value of the original Hamiltonian is given by

$$\langle H \rangle_p = -NK_0 - NK\partial_K f . \quad (48)$$

After some algebra, one can massage this into the form

$$\langle H \rangle_p = -NK_0 - 2NK\eta(K), \quad (49)$$

where

$$\eta(K) = \frac{1}{\pi}\int_0^\pi d\phi \, \frac{\tanh(2K)}{\sqrt{1 - 4\tanh^2(2K)\,\mathrm{sech}^2(2K)\cos^2\phi}} \\ \times \left(1 - \frac{2\,\mathrm{sech}^2(2K)\cos^2\phi}{1 + \sqrt{1 - 4\tanh^2(2K)\,\mathrm{sech}^2(2K)\cos^2\phi}}\right). \quad (50)$$

Meanwhile, the expectation value of the Hamiltonian after one decimation step is

$$\langle H' \rangle_{p'} = -\frac{N}{2}K_0' - NK'\eta(K'). \quad (51)$$

Therefore, the normalized relative entropy after $n$ decimation steps is given by

$$s^{(n)} = \frac{1}{\ln 2}\left(K_0 + 2K\eta(K) - \frac{1}{2^n}K_0^{(n)} - \frac{1}{2^{n-1}}K^{(n)}\eta\left(K^{(n)}\right)\right) + 1 - \frac{1}{2^n}. \quad (52)$$

As it should, $K_0$ once again drops out of this expression. The plot of this normalized relative entropy as a function of the decimation step for various values of the nearest-neighbor interaction coupling constant $K$ within the $\alpha$-extended Maris-Kadanoff approach (45) is shown in fig. 3 and 4. Here, we have chosen $\alpha = \alpha_*$ in (46) so that the value of the critical coupling $K_c$ matches the exact result given in (44).

As expected from our analysis above, we observe a qualitative difference in behavior for $K$ above vs. below the critical value $K_c$. For the small values of $K$ in fig. 3, the behavior is the same that we observed in the 1d Ising model: the relative entropy monotonically increases to some asymptotic value that depends on the coupling strength. For the large values of $K$ in fig. 4 however, the breakdown of the approximation already becomes apparent when $K$

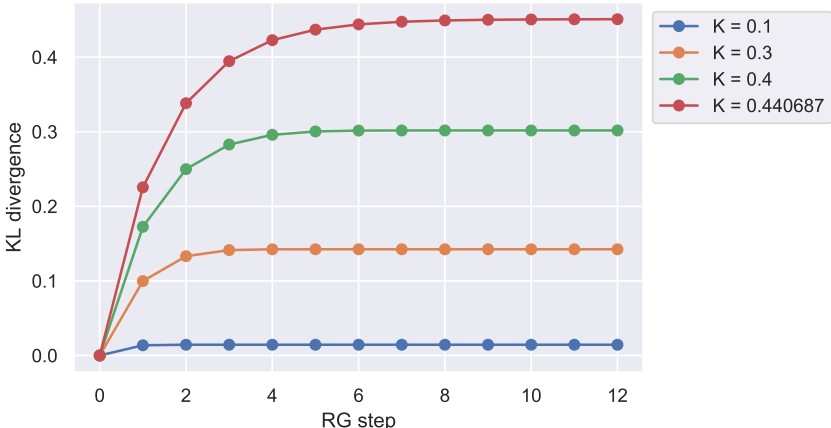

Figure 3: The normalized relative entropy of the 2d classical isotropic Ising model as a function of decimation step for values of the nearest-neighbor coupling $K < K_c$, within the $\alpha$-extended Maris-Kadanoff approach (45) with $\alpha = \alpha_*$ in (46). The behavior nicely reproduces our intuition from the 1d Ising model above.

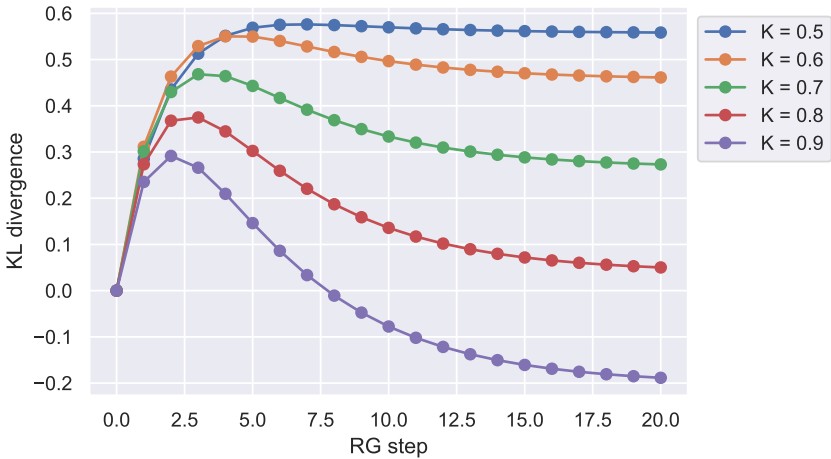

Figure 4: Same as fig. 3, but with $K > K_c$. The non-monotonic decrease and negative values signal a clear breakdown of the approximation, as discussed in the main text.

is only slightly above the critical point: in the $K = 0.5$ curve, the relative entropy slightly overshoots its asymptote before approaching it from above. This behavior becomes more and more pronounced as we increase $K$ even further, until eventually the approximation is so poor that the normalized relative entropy becomes negative. We emphasize that negativity in the *normalized* relative entropy is indeed pathological, as the change in dimensionality responsible for the (non-pathological) negativity in the unnormalized relative entropy has already been taken into account. See also [31] for further discussion of this behavior in the context of the closely related KL divergence considered therein.

Since the approximation breaks down above $K_c$, we can only reliably study the asymptotic value of the normalized relative entropy for $K \leq K_c$. In the small-$K$ limit, one finds

$$s^{(n)} = \frac{K^2}{\ln 2} + O(K^4), \qquad n \geq 1.$$ (53)

We take $s^{(\infty)}$ to be well-approximated by $s^{(12)}$, and plot this approximation on top of the numerical results in fig. 5, showing good agreement for small values of $K$.

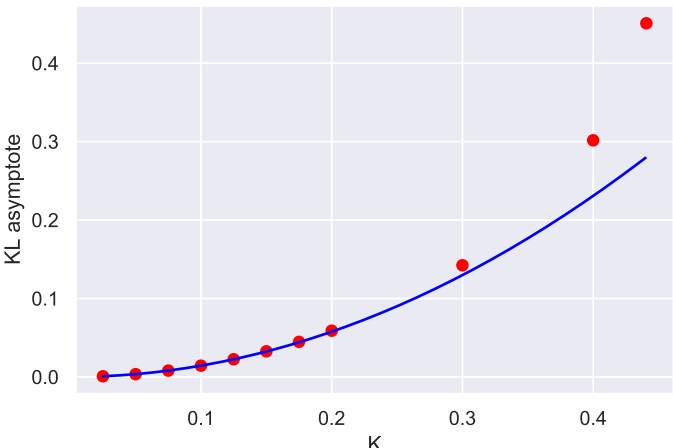

Figure 5: The asymptote of the normalized relative entropy of the 2d classical isotropic Ising model as a function of the nearest-neighbor coupling $K$ in the range $K \leq K_c$. The solid line is the function (53), while the red dots are the numerical data.

Having completed our analysis of the relative entropy in these simple spin models, let us now turn our attention to deep neural networks, to see whether the analogy with RG – with subsequent layers playing the role of successive RG steps – extends to the behavior of the KL divergence in these systems.

## 3 Relative entropy in deep neural nets

In this section, we will obtain an explicit expression for the KL divergence as a function of depth for a simple feedforward random network. The network consists of neurons $z_i^\ell$ arranged in layers $\ell \in \{0, \ldots, L-1\}$, with $i \in \{0, \ldots, N^\ell - 1\}$ neurons per layer. The output value of each neuron is called the pre-activation, and is given by

$$z_i^\ell = \sum_j W_{ij}^\ell y_j^{\ell-1} + b_i^\ell \,, \tag{54}$$

where $W$ is an $N^\ell \times N^{\ell-1}$ matrix of connection weights, $b$ is the bias, and $y_j^{\ell-1} = \phi(z_j^\ell)$ is some non-linear activation function of the neurons in the previous layer; here, as in [23,24], we shall limit ourselves to the common choice $\phi(z) = \tanh(z)$. Note that in this expression, the index $j$ runs over all neurons in the previous layer, but that there are no connections between neurons in the same layer; such networks are called fully-connected. In random networks, the weights and biases are taken to be independent and identically distributed (i.i.d.) as $W_{ij}^\ell \sim \mathcal{N}(0, \sigma_w^2/N^{\ell-1})$, $b_i^\ell \sim \mathcal{N}(0, \sigma_b^2)$, where the rescaling of the variance $\sigma_w^2$ ensures that the pre-activations remain order 1 regardless of layer width[13]. It is then straightforward to show that for any fixed set of activations $y$, the distribution of neurons on each layer is also Gaussian,

$$z_i^\ell \sim \mathcal{N}(0, \sigma_z^2)\,, \qquad \text{with} \qquad \sigma_z^2 = \frac{\sigma_w^2}{N^{\ell-1}} \sum_j (y_j^{\ell-1})^2 + \sigma_b^2\,. \tag{55}$$

---

[13]While other initialization schemes show better performance on standard benchmark tasks, Gaussian initialization makes the distributions immediately tractable without having to appeal to the central limit theorem. As will be discussed below, this does not fundamentally change the structural analogy with RG under study.

In the large-$N$ limit, this becomes the variance of the distribution of neurons for the entire layer $\ell$; following [23,24], we shall denote this by $q^\ell := \sigma_z^2$. Replacing the sum by an integral, we have the following recursion relation for the variance:

$$q^\ell = \sigma_w^2 \int \mathcal{D}z \, \phi\left(\sqrt{q^{\ell-1}} z\right)^2 + \sigma_b^2 \,, \tag{56}$$

where $\mathcal{D}z = (2\pi)^{-1/2} e^{-z^2/2}$ is the standard Gaussian measure.

Now, as mentioned in the introduction, the critical point – in the 2d phase space parametrized by $\sigma_w^2$, $\sigma_b^2$ – is characterized by a divergence in some correlation length. To find this, [23] consider the two-point correlator (i.e., the covariance matrix) between two different inputs to the same network, denoted $x_a = \{x_{i,a}, \ldots, x_{N,a}\}$ (that is, $y_a^0 = x_a$). The correlation between pre-activations is then

$$\langle z_{i,a}^\ell z_{i,b}^\ell \rangle = \frac{\sigma_w^2}{N^{\ell-1}} \sum_j y_{j,a}^{\ell-1} y_{j,b}^{\ell-1} + \sigma_b^2 , \tag{57}$$

which is denoted $q_{ab}^\ell := \text{cov}(z_a^\ell, z_b^\ell)$. As above, one can then obtain a recursion relation for the covariance in the large-$N$ limit [23]

$$q_{ab}^\ell = \sigma_w^2 \int \mathcal{D}z_1 \mathcal{D}z_2 \phi(z_a) \phi(z_b) + \sigma_b^2 \,, \tag{58}$$

where the standard normal variables $z_1$, $z_2$ are related to $z_a$, $z_b$ as

$$z_a = \sqrt{q_a^{\ell-1}} z_1, \qquad z_b = \sqrt{q_b^{\ell-1}} \left( \rho^{\ell-1} z_1 + \sqrt{1 - (\rho^{\ell-1})^2} z_2 \right), \tag{59}$$

and $\rho^\ell := q_{ab}^\ell / \sqrt{q_a^\ell q_b^\ell}$ is the Pearson correlation coefficient, which takes values between 0 (completely uncorrelated) and 1 (completely correlated).

Interestingly, [23] showed that both (56) and the correlation $\rho$ exhibit fixed points at particular points in the 2d phase space parametrized by $\sigma_w^2$, $\sigma_b^2$. The former is defined by

$$q^* = \sigma_w^2 \int \mathcal{D}z \, \tanh(\sqrt{q^*} z)^2 + \sigma_b^2 \,, \tag{60}$$

at which

$$\rho^\ell = \frac{\sigma_w^2}{q^*} \int \mathcal{D}z_1 \mathcal{D}z_2 \, \phi(z_a^*) \phi(z_b^*) + \frac{\sigma_b^2}{q^*} \,, \tag{61}$$

where the asterisks denote the evaluation of $z_a, z_b$ at $\rho = \rho^*$, $q = q^*$. It is easy to see that this expression admits a fixed point at $\rho^* = 1$, corresponding to perfect correlation. One can then probe the stability of this fixed point by considering

$$\left. \frac{\partial \rho^\ell}{\partial \rho^{\ell-1}} \right|_{\rho^{\ell-1} = \rho^*} = \sigma_w^2 \int \mathcal{D}z \, \phi'(\sqrt{q^*} z)^2 =: \chi_1 , \tag{62}$$

where $\phi'(x) = \partial_x \phi(x)$. In particular, if $\chi_1 < 1$, $\rho^\ell$ will be driven towards $\rho^*$, and hence the fixed point is stable. Conversely, if $\chi_1 > 1$, $\rho^\ell$ will be driven away from $\rho^*$, and the fixed point is unstable. (To see this, plot the recursion relation for the correlation as in fig. 2 of [23], and observe that if $\chi_1 < 1$, then the curve must approach the fixed point from above, so that any initial point is driven to unity; conversely, $\chi_1 > 1$ implies that the curve approaches the fixed point from below, so that any initial point will be driven to some finite value near zero). Since

the fixed point lies at $\rho^* = 1$, corresponding to perfect correlation between the two inputs, the line $\chi_1 = 1$ defines an order-to-disorder phase transition in the $(\sigma_w^2, \sigma_b^2)$ plane. For $\chi_1 > 1$, the correlation between inputs will tend to shrink, signalling a disordered or chaotic phase, while for $\chi_1 > 1$, any differences between inputs will tend to be washed out. Intuitively, both phases are bad for learning, since they inhibit the transmission of useful correlations through the network.

In physics, such a *critical point* at $\chi_1 = 1$ is characterized by a divergent correlation length. This is worked-out in [24], and denoted $\xi_c$:

$$\xi_c^{-1} = -\ln \sigma_w^2 \int \mathcal{D}z_1 \mathcal{D}z_2 \, \phi'(z_a^*)\phi'(z_b^*). \tag{63}$$

One can numerically evaluate this for a particular point $(\sigma_w^2, \sigma_b^2)$ in phase space by first finding the value of $q^*$ via (60), and then using this to obtain the corresponding value of $\rho^*$ via (61), which we overlay in fig. 6. Precisely at $\rho^* = 1$, however, the argument of the logarithm reduces to $\chi_1 = 1$, and thus the correlation length diverges at the critical point as expected.

This is an important result that goes a long way to explaining the trainability of networks of different depths, to wit: such networks are not trainable if their depth exceeds the correlation length. This implies that initializing networks near criticality will enable one to train substantially deeper networks than would otherwise be possible; see for example [25], in which this was used to train a network with an unprecedented 10,000 layers. This idea is illustrated in fig. 5 of [24], which we have reproduced for our custom networks in fig. 6. Even for our relatively crude experiments, one sees a clear fall-off in trainability on either side of the critical point. Curiously, as in [24,25], the fall-off does not correspond to the correlation length itself, but to some $\sigma_w^2$-dependent function thereof. We suspect that this is due to finite-width effects, which give subleading corrections to the large-$N$ "mean field" analysis reviewed above [10,60].

Due to both the practical and theoretical importance of criticality in deep neural networks, it is worth asking whether additional probes of information flow through these networks might offer complementary insights into this behavior. In light of the aforementioned parallels between the structure of feedforward networks and real-space RG, we thus turn to an evaluation of the Kullback-Leibler divergence as a function of depth. While depth is analogous to RG step in our Ising calculations above, the computation in the present case is complicated by the recursive nature of the map from layer to layer. That is, the set of spins at step $\ell$ were obtained by simply marginalizing over some subset of spins at step $\ell-1$, while the neurons in layer $\ell$ are related to those in layer $\ell-1$ by some non-linear function (54). Nonetheless, the simplicity of these random networks in the large-$N$ limit (i.e., Gaussians) enables us to obtain an explicit expression for the KL divergence through the network, which can be evaluated numerically. To our knowledge, this is the first explicit calculation of relative entropy as a function of depth to have appeared in the literature.

## 3.1 Explicitly computing the KL divergence

To warm up, let $p(z)$, $q(z)$ denote the normal distributions, with potentially different means and standard deviations, of some continuous random variable $z$, i.e.,

$$p(z) = \frac{1}{\sqrt{2\pi\sigma_p^2}} e^{-\frac{1}{2}\left(\frac{z-\mu_p}{\sigma_p}\right)^2}, \tag{64}$$

and similarly for $q$. The relative entropy between these distributions – with $q$ interpreted as the approximation to or description of the reference distribution $p$ – is

$$D(p||q) = \int dz \, p(z) \ln \frac{p(z)}{q(z)} = \langle \ln p(z) \rangle_p - \langle \ln q(z) \rangle_p = -S(p) - \langle \ln q(z) \rangle_p, \tag{65}$$

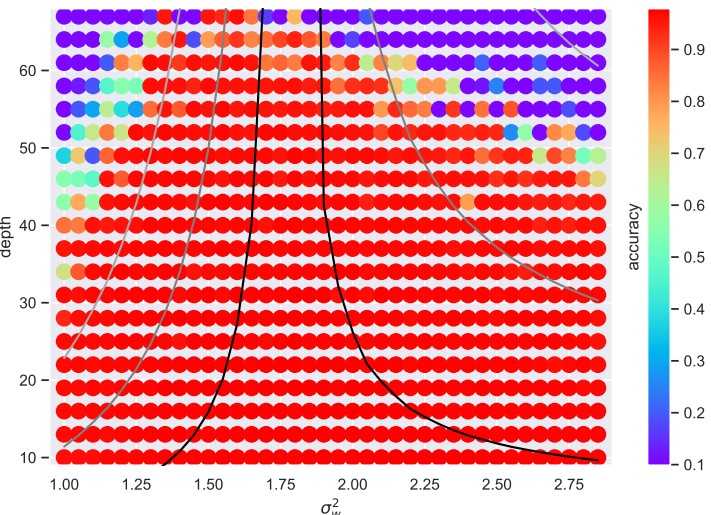

Figure 6: Classification accuracy on MNIST after 15 epochs of training as a function of $\sigma_w^2$ and depth for $\sigma_b^2 = 0.05$, for which the critical point $\chi_1 = 1$ lies at $\sigma_w^2 \approx 1.76$ (at $N \to \infty$). The theoretical value of 1, $\pi$, and $2\pi$ times the correlation length are overlaid in black, grey, and light grey, respectively. Note that we can train deeper networks closer to criticality; the maximum trainable depth falls off the further we move into the ordered ($\sigma_w^2 \lesssim 1.76$) or chaotic ($\sigma_w^2 \gtrsim 1.76$) phase. This serves as a sanity check that our networks exhibit the same qualitative behavior observed in [24], cf. fig. 5 therein, which depicts a more thorough exploration (because this is the best we could do before hitting Colab's resource limits).

where $S(p)$ is the entropy

$$S(p) = -\langle \ln p(z) \rangle_p = \frac{1}{2} \ln 2\pi e \sigma_p^2 . \tag{66}$$

The remaining expectation value is readily evaluated:

$$\langle \ln q(z) \rangle_p = -\frac{1}{2} \left\langle \left( \frac{z - \mu_q}{\sigma_q} \right)^2 \right\rangle_p - \frac{1}{2} \ln 2\pi \sigma_q^2 . \tag{67}$$

Continuing with the first term, we have

$$\langle (z - \mu_q)^2 \rangle_p = \langle z^2 \rangle_p - 2\mu_q \langle z \rangle_p + \mu_q^2 = \sigma_p^2 + (\mu_p - \mu_q)^2 . \tag{68}$$

Thus we obtain

$$\langle \ln q(z) \rangle_p = -\frac{1}{2\sigma_q^2} \left[ \sigma_p^2 + (\mu_p - \mu_q)^2 \right] - \frac{1}{2} \ln 2\pi \sigma_q^2 . \tag{69}$$

Substituting this and (66) into (65), we obtain

$$D(p||q) = -\frac{1}{2} \ln 2\pi e \sigma_p^2 + \frac{1}{2} \ln 2\pi \sigma_q^2 + \frac{1}{2\sigma_q^2} \left[ \sigma_p^2 + (\mu_p - \mu_q)^2 \right] . \tag{70}$$

Note that when $q = p$, the last term reduces to $1/2$ and combines with the second term to yield the entropy (66), in which case $S(p||q) = 0$, as expected.

The above result is straightforwardly generalized to the multidimensional case

$$p(\mathbf{z}) = \frac{1}{\sqrt{(2\pi)^N |\Sigma_p|}} \, e^{-\frac{1}{2}(\mathbf{z}-\mu_p)^{\mathrm{T}} \Sigma_p^{-1}(\mathbf{z}-\mu_p)} \,, \tag{71}$$

where $\Sigma_p$ is the $N \times N$ covariance matrix, and $\mathbf{z}, \mu$ are column $N$-vectors, and similarly for $q(\mathbf{z})$. Fortunately, we are here concerned with the case in which $\Sigma_p = \mathrm{diag}(\sigma_1^2, \sigma_2^2, \ldots, \sigma_N^2)$, which simplifies the computations.

The first term in (65) is just the entropy of a multivariate Gaussian, and can be computed in generality:

$$S(p(\mathbf{z})) = \frac{1}{2} \ln |2\pi e \Sigma_p| \,, \tag{72}$$

where $|\cdot|$ denotes the determinant (this is easily obtained via the so-called "trace trick"). As for the second term, since *both* $\Sigma_q$ and $\Sigma_p$ are diagonal, the expectation value in (67) becomes

$$\begin{aligned}
\langle (\mathbf{z}-\mu_q)^{\mathrm{T}} \Sigma_q^{-1}(\mathbf{z}-\mu_q) \rangle_p &= \sum_{i=1}^{N} [\Sigma_q^{-1}]_{ii} \langle (z_i - \mu_{q,i})^2 \rangle_p \\
&= \sum_{i=1}^{N} [\Sigma_q^{-1}]_{ii} \left( [\Sigma_p]_{ii} + (\mu_{p,i} - \mu_{q,i})^2 \right) \\
&= \mathrm{tr}\, \Sigma_q^{-1} \left[ \Sigma_p + (\mu_p - \mu_q)^2 \right] \,,
\end{aligned} \tag{73}$$

where $[\Sigma]_{ii}$ is the diagonal element of $\Sigma$. In going to the second line, we have used the factorization of the multivariate Gaussian (i.e., diagonality of $\Sigma_p^{-1}$), and the fact that the integrals over all $z_j \neq z_i$ evaluate to unity, leaving a sum of terms of the form (68). (Note also that the trace acts on everything to the right, not just $\Sigma_q^{-1}$, we are simply suppressing a surplus of brackets). Thus in place of (69), we have

$$\langle q(\mathbf{z}) \rangle_p = -\frac{1}{2} \mathrm{tr}\, \Sigma_q^{-1} \left[ \Sigma_p + (\mu_p - \mu_q)^2 \right] - \frac{1}{2} \ln |2\pi \Sigma_q| \,. \tag{74}$$

The KL divergence for the multivariate case, with diagonal covariance matrices, is therefore

$$D(p(\mathbf{z})||q(\mathbf{z})) = -\frac{1}{2} \ln |2\pi e \Sigma_p| + \frac{1}{2} \ln |2\pi \Sigma_q| + \frac{1}{2} \mathrm{tr}\, \Sigma_q^{-1} \left[ \Sigma_p + (\mu_p - \mu_q)^2 \right]. \tag{75}$$

Note that when $q = p$, the last term reduces to $N/2$, so that we again recover $D(p||p) = 0$.

Now we wish to extend this result to a hierarchical network, in which the reference distribution $p(\mathbf{z})$ describes the lowest (input) layer, and the approximate or model distribution $q(\mathbf{z})$ describes any higher layer; i.e., we wish to compute $D(p(\mathbf{z}^\ell)||q(\mathbf{z}^{\ell+m}))$ where $m \in [0, L-1]$[14]. When $m = 0$ we recover (75) with $q = p$, while at the other extreme, $m = L-1$, we have the relative entropy between the input and output layers for an $L$ layer network. As explained above, this is complicated because $\mathbf{z}^{\ell+1}$ is a recursive function of $\mathbf{z}^\ell$ given by (54), which in vector notation reads

$$\mathbf{z}^{\ell+1} = W^{\ell+1} \phi(\mathbf{z}^\ell) + \mathbf{b}^{\ell+1} \,, \tag{76}$$

where $\phi(\mathbf{z}^\ell)$ is understood as a (column) vector whose elements are the nonlinear activation function $\phi$ acting on the individual neurons in the previous layer, and $W$ and $\mathbf{b}$ are the $N^{\ell+1} \times N^\ell$ matrix of weights and $N^{\ell+1}$ column vector of biases, respectively.

---

[14] Note that while this isn't strictly an RG, and hence we could optionally reverse the order of $p$ and $q$, we have chosen this ordering so that expectation values are always computed with respect to the "UV" distribution, for the reasons explained in the previous section.

Since the covariance matrices are still diagonal, the only change to the above computation is in the calculation of the expectation values $\left\langle \left( z_i^{\ell+m} \right)^2 \right\rangle_{p(\mathbf{z}^\ell)}$, $\left\langle z_i^{\ell+m} \right\rangle_{p(\mathbf{z}^\ell)}$. Let us start with $m=1$, and try to obtain some useful recursion relations. We have

$$\left\langle \left( z_i^{\ell+1} \right)^2 \right\rangle = \sum_{j,k} W_{ij}^{\ell+1} W_{ik}^{\ell+1} \langle \phi(z_j^\ell)\phi(z_k^\ell)\rangle_p + 2\sum_j W_{ij}^{\ell+1} b_i^{\ell+1} \langle \phi(z_j^\ell)\rangle + \left( b_i^{\ell+1} \right)^2 \tag{77}$$

and

$$\left\langle z_i^{\ell+1} \right\rangle = \sum_j W_{ij}^{\ell+1} \langle \phi(z_j^\ell)\rangle + b_i^{\ell+1} . \tag{78}$$

Taking $\phi(z) = \tanh(z)$, and using the factorization $p(\mathbf{z}) = \prod_i p(z_i)$ (i.e., diagonality of $\Sigma_p$ as before), the remaining expectation values become

$$\langle \phi(z_j^\ell)\rangle_{p(\mathbf{z}^\ell)} = \frac{1}{\sigma_{p,j}^\ell \sqrt{2\pi}} \int dz_j^\ell \tanh(z_j^\ell) \, e^{-\frac{1}{2}\left(\frac{z_j^\ell - \mu_{p,j}^\ell}{\sigma_{p,j}^\ell}\right)^2} =: f_j^0 \tag{79}$$

and

$$\langle \phi(z_j^\ell)\phi(z_k^\ell)\rangle_{p(\mathbf{z}^\ell)} = \frac{1}{2\pi \sigma_{p,j}^\ell \sigma_{p,k}^\ell} \int dz_j^\ell dz_k^\ell \tanh(z_j^\ell)\tanh(z_k^\ell)$$
$$\times e^{-\frac{1}{2}\left(\frac{z_j^\ell - \mu_{p,j}^\ell}{\sigma_{p,j}^\ell}\right)^2 - \frac{1}{2}\left(\frac{z_k^\ell - \mu_{p,k}^\ell}{\sigma_{p,k}^\ell}\right)^2} =: f_{j,k}^0 , \tag{80}$$

which unfortunately must be evaluated numerically; we have denoted them by $f_j^0$ and $f_{j,k}^0$ for compactness. With these in hand, (73) becomes

$$\left\langle \left( \mathbf{z}^{\ell+1} - \boldsymbol{\mu}_q^{\ell+1} \right)^{\mathrm{T}} \Sigma_q^{\ell+1} \left( \mathbf{z}^{\ell+1} - \boldsymbol{\mu}_q^{\ell+1} \right) \right\rangle_p = \sum_{i=1}^{N^{\ell+1}} \left[ \left( \Sigma_q^{\ell+1} \right)^{-1} \right]_{ii} \left\langle \left( z_i^{\ell+1} - \mu_{q,i}^{\ell+1} \right)^2 \right\rangle_p$$

$$= \sum_{i=1}^{N^{\ell+1}} \left[ \left( \Sigma_q^{\ell+1} \right)^{-1} \right]_{ii} \Big[ \sum_{j,k} W_{ij}^{\ell+1} W_{ik}^{\ell+1} f_{j,k}^0 + 2\sum_j W_{ij}^{\ell+1} b_i^{\ell+1} f_j^0 + \left( b_i^{\ell+1} \right)^2$$

$$- 2\mu_{q,i}^{\ell+1} \left( \sum_j W_{ij}^{\ell+1} f_j^0 + b_i^{\ell+1} \right) + \left( \mu_{q,i}^{\ell+1} \right)^2 \Big]$$

$$= \sum_{i=1}^{N^{\ell+1}} \left[ \left( \Sigma_q^{\ell+1} \right)^{-1} \right]_{ii} \Big[ \sum_{j,k} W_{ij}^{\ell+1} W_{ik}^{\ell+1} f_{j,k}^0 + 2\sum_j W_{ij}^{\ell+1} \left( b_i^{\ell+1} - \mu_{q,i}^{\ell+1} \right) f_j^0 + \left( b_i^{\ell+1} - \mu_{q,i}^{\ell+1} \right)^2 \Big]$$

$$=: \langle Q^{\ell+1} \rangle_p , \tag{81}$$

where the identification in the last line has been introduced for shorthand. Collecting results, we obtain

$$D(p(\mathbf{z}^\ell)\|q(\mathbf{z}^{\ell+1})) = -S(p(\mathbf{z}^\ell)) - \langle \ln q(\mathbf{z}^{\ell+1})\rangle_p$$
$$= -\frac{1}{2} \ln |2\pi e \Sigma_p^\ell| + \frac{1}{2} \ln |2\pi \Sigma_q^{\ell+1}| + \frac{1}{2} \langle Q^{\ell+1} \rangle_p , \tag{82}$$

where the first, second, and third terms are precisely analogous to those in (75). Again, we recover the null result when $\phi(z^\ell) = z^{\ell+1}$, $W=1$, and $b=0$, corresponding to $q=p$.

Now, since all the recursion is contained in the functions $f_j^0$, $f_{j,k}^0$, we can easily write – at least formally – the expression for arbitrary $m$:

$$\begin{aligned}
D(p(\mathbf{z}^\ell)\|q(\mathbf{z}^{\ell+m})) &= -S(p(\mathbf{z}^\ell)) - \langle \ln q(\mathbf{z}^{\ell+m}) \rangle_p \\
&= -\frac{1}{2}\ln|2\pi e \Sigma_p^\ell| + \frac{1}{2}\ln|2\pi \Sigma_q^{\ell+m}| + \frac{1}{2}\langle Q^{\ell+m} \rangle_p\,,
\end{aligned} \tag{83}$$

where

$$\begin{aligned}
\langle Q^{\ell+m} \rangle_p &:= \sum_{i=1}^{N^{\ell+m}} \left[ \left(\Sigma_q^{\ell+m}\right)^{-1} \right]_{ii} \Big[ \sum_{j,k} W_{ij}^{\ell+m} W_{ik}^{\ell+m} f_{j,k}^{m-1} \\
&\quad + 2\sum_j W_{ij}^{\ell+m}\left(b_i^{\ell+m} - \mu_{q,i}^{\ell+m}\right) f_j^{m-1} + \left(b_i^{\ell+m} - \mu_{q,i}^{\ell+m}\right)^2 \Big]\,,
\end{aligned} \tag{84}$$

and the recursive functions to be computed numerically are

$$\begin{aligned}
f_j^{m-1} &:= \langle \phi(z_j^{\ell+m-1}) \rangle_{p(\mathbf{z}^\ell)} \\
&= \frac{1}{\sqrt{\left|2\pi\Sigma_p^\ell\right|}} \int d\mathbf{z}^\ell \, \tanh\left(z_j^{\ell+m-1}\right) e^{-\frac{1}{2}\left(\mathbf{z}^\ell - \mu_p^\ell\right)^{\mathrm{T}}\left(\Sigma_p^\ell\right)^{-1}\left(\mathbf{z}^\ell - \mu_p^\ell\right)}\,,
\end{aligned} \tag{85}$$

$$\begin{aligned}
f_{j,k}^{m-1} &:= \langle \phi(z_j^{\ell+m-1})\phi(z_k^{\ell+m-1}) \rangle_{p(\mathbf{z}^\ell)} \\
&= \frac{1}{\sqrt{\left|2\pi\Sigma_p^\ell\right|}} \int d\mathbf{z}^\ell \, \tanh(z_j^{\ell+m-1})\tanh(z_k^{\ell+m-1}) e^{-\frac{1}{2}\left(\mathbf{z}^\ell - \mu_p^\ell\right)^{\mathrm{T}}\left(\Sigma_p^\ell\right)^{-1}\left(\mathbf{z}^\ell - \mu_p^\ell\right)}\,,
\end{aligned} \tag{86}$$

where $d\mathbf{z}^\ell := \prod_r dz_r^\ell$. Note that unlike $f_j^0$ and $f_{j,k}^0$ above, we retain the integration over all vector components for $m > 1$, since all $z_k^\ell$ will appear in the tanh in the integrand via the recursion relation (76). For $m = 1$, the integrals over $z_r^\ell$ for $r \neq j, k$ evaluate to 1, and we recover $f_j^0$, $f_{j,k}^0$ above. While we've written these expressions in full generality, as mentioned at the beginning of this section we intend to take $p(\mathbf{z}^\ell)$ to be the input layer, in which case we set $\ell = 0$; then $m$ simply indexes the depth through the network.

Note that the recursive contributions are the same for all $i$, and only depend on the indices $j, k$. Therefore in practice, it is more computationally efficient to change the order of summation. We can also use the fact that, in this approximation, $\Sigma_q = \sigma_q^2 \mathrm{diag}(1,\ldots,1)$ to move the covariance outside the sum. Lastly, we shall set the reference layer $\ell = 0$, so we have simply

$$\sigma_q^2 \langle Q^{\ell+m} \rangle_p := \sum_{j,k}^{N^{m-1}} f_{j,k}^{m-1} \sum_i^{N^m} W_{ij}^m W_{ik}^m + 2\sum_j^{N^{m-1}} f_j^{m-1} \sum_i^{N^m} W_{ij}^m\left(b_i^m - \mu_{q,i}^m\right) + \sum_i^{N^m}\left(b_i^m - \mu_{q,i}^m\right)^2\,. \tag{87}$$

## 3.2 Implementation and results

Since our focus in this work is a theoretical analysis of the KL divergence rather than state-of-the-art machine learning performance, we will employ a basic feedforward random network trained on the standard MNIST dataset [61] as well as CIFAR-10 (converted to greyscale), with no optimizers or other bells and whistles[15]. The key feature is the random initialization and relatively large width (784 for the $28 \times 28$ pixels comprising a given MNIST image, or 1024 for CIFAR-10 after converting to greyscale), for which the bulk layers remain approximately Gaussian.

---

[15]The code to reproduce all results is available at https://github.com/ro-jefferson/entropy_dnn.

Turning now to the practicalities of evaluating the KL divergence (83), the only really annoying bit is the recursive computation of the integrals, i.e., of $\tanh(z_j^{\ell+m-1}) = \phi(z_j^{\ell+m-1})$:

$$
\begin{aligned}
\phi(z_j^{\ell+m-1}) &= \phi\left(\sum_k W_{jk}^{\ell+m-1} \phi(z_j^{\ell+m-2}) + b_j^{\ell+m-1}\right) \\
&= \phi\left(\sum_k W_{jk}^{\ell+m-1} \phi\left(\sum_r W_{kr}^{\ell+m-2} \phi(z_r^{\ell+m-3}) + b_k^{\ell+m-2}\right) + b_j^{\ell+m-1}\right) \\
&\vdots \\
&= \phi\left(\sum_k W_{jk}^{\ell+m-1} \phi\left(\cdots \sum_s W_{ts}^{\ell+1} \phi(z_s^{\ell}) + b_t^{\ell+1} \cdots\right) + b_j^{\ell+m-1}\right),
\end{aligned}
\tag{88}
$$

where the ellipses in the last line are meant to indicate the recursive substitution of $z^{\ell+1}$ as per (76). Naïvely, it seems we require a recursive function that computes $\phi(z_i^{\ell+m-1})$ in terms of $z_j^{\ell}$. Setting the reference layer $\ell = 0$, this should return, for sequential values of $m$ (and using Einstein summation notation),

$$
\begin{aligned}
m=1 \,,\; f_i^0 \;:\;\; & \phi(z_i^0) = \tanh(z_i^0) \,, \\
m=2 \,,\; f_i^1 \;:\;\; & \phi(z_i^1) = \phi\left(W_{ij}^1 \phi(z_j^0) + b_i^1\right) = \phi\left(W_{i0}^1 \phi(z_0^0) + \ldots + W_{in_0}^1 \phi(z_n^0) + b_i^1\right) \,, \\
m=3 \,,\; f_i^2 \;:\;\; & \phi(z_i^2) = \phi\left(W_{ij}^2 \phi\left(W_{jk}^1 \phi(z_k^0) + b_j^1\right) + b_i^2\right) \\
& = \phi\left(W_{i0}^2 \phi\left(W_{0k}^1 \phi(z_k^0) + b_0^1\right) + \ldots + W_{in_1}^2 \phi\left(W_{n_1 k}^1 \phi(z_k^0) + b_{n_1}^1\right) + b_i^2\right) \\
& = \phi\left(W_{i0}^2 \phi\left(W_{00}^1 \phi(z_0^0) + \ldots + W_{0n_0}^1 \phi(z_{n_0}^0) + b_0^1\right)\right. \\
& \quad + \ldots + \\
& \quad \left. W_{in_1}^2 \phi\left(W_{n_1 0}^1 \phi(z_0^0) + \ldots + W_{n_1 n_0}^1 \phi(z_{n_0}^0) + b_{n_1}^1\right) + b_i^2\right),
\end{aligned}
\tag{89}
$$

and so on, where we have refrained from substituting in tanh in all but the first case for compactness. Note however that from $m = 3$ onwards, there is no new $z$-dependence: we have already summed-over all free indices in the weight matrices that couple to $\mathbf{z}^0$ by the time we reach $m=2$. (This is easy to see visually by sketching out the architecture of the network: in order to compute the activation for a neuron in layer 2, we must sum over all the neurons in layer 1 (one index), and then sum over all the neurons in layer 0 for each of them (two indices)). Indeed, it is computationally very inefficient to re-compute the activations $\phi(\mathbf{z}^m)$ of a given layer every time we step through the recursive tower. Rather, given sufficient RAM, the following procedure provides a speed-up of about 6 orders of magnitude in our experiments on a humble desktop machine.

We fix the reference layer to be $\ell = 0$, and denote the target layer by $m > \ell$ as above. The high-dimensional nature of the integrals (85), (86) lend themselves to Monte Carlo methods, for which we require $n_{\text{MC}}$ samples. The algorithm then proceeds as follows:

1. Starting at $m = 1$, compute $n_{\text{MC}}$ samples of the vector of activations $\phi(\mathbf{z}^0) = \{\phi(z_i^0)\}$, drawn from the reference layer (see appendix B).

2. At $m=2$, multiply each element of this vector by the appropriate weight, and sum over $j$ to obtain $n_{\text{MC}}$ samples of $W_{ij}^1 \phi(z_j^0)$. After adding the bias $b_i^1$, taking the tanh, and repeating for all $i$, we obtain $n_{\text{MC}}$ samples of the vector $\phi(\mathbf{z}^1) \equiv \{\phi(z_i^1)\} = \{\tanh(W_{ij}^1 \phi(z_j^0) + b_i^1)\}$.

3. Iterate the previous step to obtain $n_{\text{MC}}$ samples of $\phi(\mathbf{z}^{m-1})$ for $m \geq 3$.

4. Use the $n_{\mathrm{MC}}$ samples of $\phi(\mathbf{z}^{m-1})$ from each layer $m$ to evaluate $f_j^{m-1}$, $f_{j,k}^{m-1}$ via Monte Carlo integration (see appendix B).

5. Substitute these into (87) to obtain the contribution $\langle Q^m \rangle$.

6. Add this to the first two terms of the KL divergence (83), which are trivial to compute given the (factorized) covariance matrix for the reference $\ell = 0$ and target $\ell = m$ layers.

The results for a network with 30 layers (so as to lie well-within the trainable regime determined in fig. 6) and five different points in parameter space are shown in fig. 7. Several comments are in order. First, we observe a close parallel with the behavior of the KL divergence in both the 1d and 2d Ising models under real-space RG, cf. figs. 1 and 3. The KL divergence sharply and briefly increases, and then asymptotes to a value which depends on the network's location in phase space; here, this is controlled by $\sigma_w^2$ (for fixed $\sigma_b^2$), while in the Ising models the asymptote is controlled by the coupling strength. In both cases, the quantity of information ceases to evolve once the KL divergence reaches its asymptotic value.

This raises several questions. As long as the network is within the trainable regime in fig. 6, we observed no marked difference in classification accuracy between networks with only a few layers and those lying well into the asymptotic region. While one might be tempted to interpret this as evidence that the additional layers are unnecessary, this may simply be a coincidence: the fact that the total information content no longer changes does *not* imply that the information is no longer being processed. The KL divergence is a relatively coarse-grained probe, the inputs to which are, in this case, ultimately just the parameters of the Gaussian characterizing each layer. Even in these relatively small networks with around 22,000 neurons, there are many, many ways to rearrange the individual weights (the fine-grained state of the network) without changing the Gaussian (the coarse-grained state). Therefore, it seems that more fine-grained probes will be necessary if one wishes to usefully quantify the information content of a given layer.

A related question is the value of the asymptote itself. At least in the 1d Ising model, we could understand this in terms of the total information in the fine-grained (UV) system, where – absent the normalization factor – the limit of zero coupling leaves us with $n \ln 2$ bits for a system of $n$ spins. For these Gaussian networks, the analogous quantity is the entropy of the initial layer, $\frac{1}{2} \ln |2\pi e \Sigma_{p(\mathbf{z}^0)}|$, which does *not* correspond to the asymptotic value for any choice of parameters we explored. Furthermore, as discussed in the introduction, there is a structure vs. dynamics distinction that must be kept in mind when drawing parallels between deep neural networks and the renormalization group. As shown in fig. 7, we observe no qualitative difference in the value of the KL asymptote before vs. after training. On the one hand, we shouldn't be too surprised by this insofar as the change in the KL divergence ultimately corresponds to a Gaussian changing its width. Said differently, RG captures the structural relationship between subsequent layers in the form of effective couplings that arise under coarse-graining, which depend on the weights (see [9] for a simple example). During training, the numerical values of the weights will change, but the form of the hierarchical relationship (that is, the formal expression for the effective couplings) will not. Nonetheless, it may be interesting to quantitatively disentangle the contributions to the entropy from the structure vs. the training dynamics; this would however require more exhaustive numerics to elucidate the role of initialization, since this sets the baseline curve against which the trained curve may be compared in fig. 7.

Another notable feature is the apparent insensitivity of the relative entropy to the phase structure of the system. The critical point for the networks in fig. 7 lies at approximately $(\sigma_b^2, \sigma_w^2) = (0.05, 1.76)$, which is between the blue and orange curves in the figure. One sees no qualitative difference in the behavior of the KL divergence as a function of depth between the two phases. For MNIST, we observe that the value of the asymptote seems to grow with

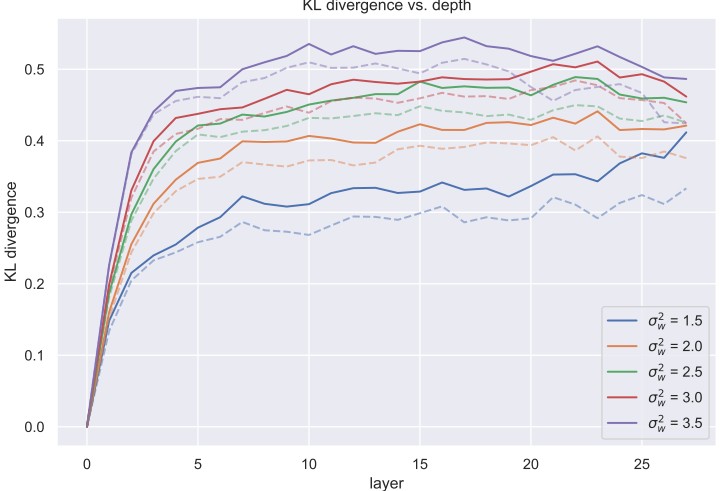

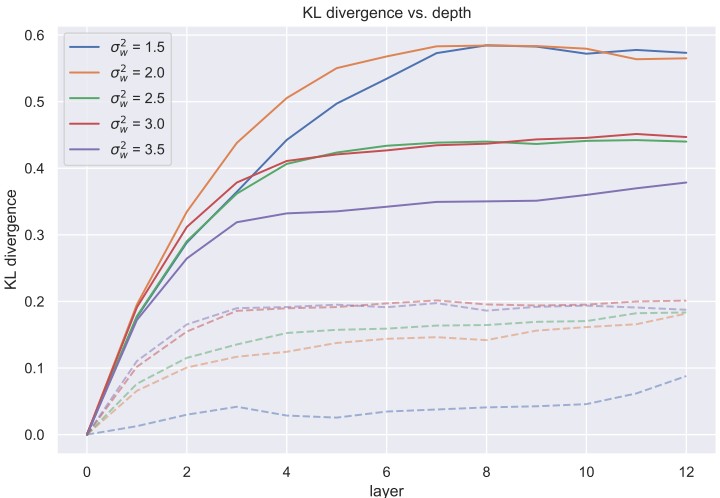

Figure 7: Plot of the KL divergence (83) before (dashed) and after (solid) training for networks with varying initializations $\sigma_w^2$ and fixed $\sigma_b^2 = 0.05$, for which the critical point corresponds to $\sigma_w^2 \approx 1.76$. Results for MNIST (top) and CIFAR-10 (bottom) are shown for networks with a fixed depth of 30 and 15 layers, respectively. The essential qualitative features – namely a sharp increase to an asymptote – are preserved through training and across different datasets.

the width of the weight distribution, but this does not appear to hold for CIFAR-10. In both cases however, one can fit the KL divergence to a function of the form

$$f(x) = a\left(1 - b^x\right) , \qquad a, b \in \mathbb{R} , \tag{90}$$

where $x$ is the depth, and $a, b$ must be determined numerically. The results for the after-training values in the case of MNIST are shown in fig. 8, where the corresponding parameters are given in table 1. While it is tempting to speculate on the $\sigma_w$-dependence of the asymptotic value $a$ (e.g., one obtains a decent fit with $a = \alpha \tanh(\beta \sigma_w^2)$ for some constants $\alpha, \beta$), we have not generated sufficient data to do so.

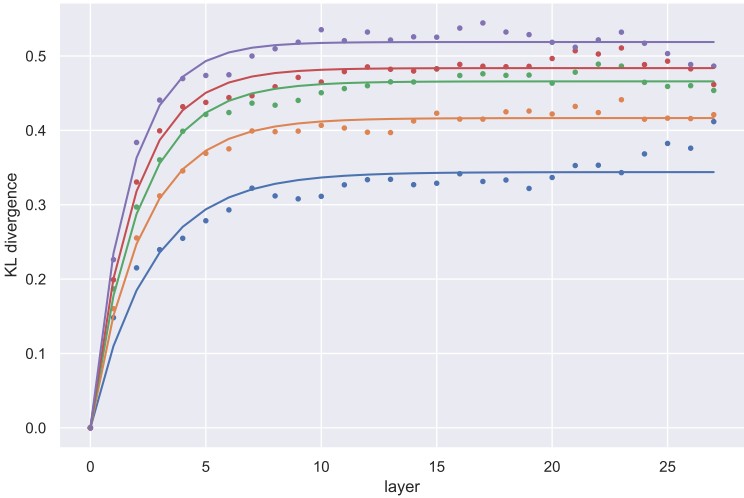

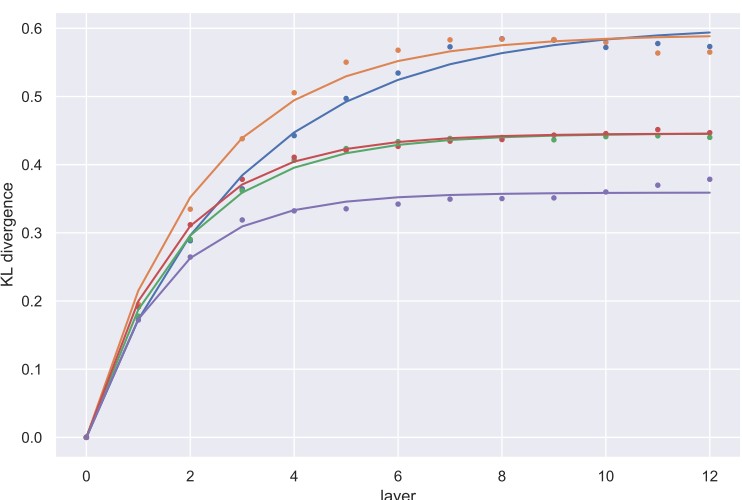

Figure 8: Fit of the after-training MNIST (top) and CIFAR-10 (bottom) data in fig. 7 to (90), with the parameter values shown in table 1. We expect one could obtain smoother results by using additional training data, or simply averaging over runs.

Table 1: Parameter values for the ansatz (90), fitted to the after-training KL data in fig. 8. Note that since $b < 1$, the second term in (90) vanishes asymptotically, so that $a$ gives the asymptotic value of the KL divergence.

<table>
<tr><td colspan="3" align="center">MNIST</td><td colspan="3" align="center">CIFAR-10</td></tr>
<tr><td>$\sigma_w^2$</td><td>$a$</td><td>$b$</td><td>$\sigma_w^2$</td><td>$a$</td><td>$b$</td></tr>
<tr><td>1.5</td><td>0.34388746</td><td>0.68008167</td><td>1.5</td><td>0.60452212</td><td>0.71402565</td></tr>
<tr><td>2.0</td><td>0.41656213</td><td>0.63731986</td><td>2.0</td><td>0.59097108</td><td>0.63554505</td></tr>
<tr><td>2.5</td><td>0.46590954</td><td>0.61860096</td><td>2.5</td><td>0.44595509</td><td>0.57956012</td></tr>
<tr><td>3.0</td><td>0.48374249</td><td>0.58489656</td><td>3.0</td><td>0.44579503</td><td>0.55155307</td></tr>
<tr><td>3.5</td><td>0.51882683</td><td>0.54771529</td><td>3.5</td><td>0.35906706</td><td>0.51715509</td></tr>
</table>

We must also comment on the issue of normalization we encountered in the Ising models, which surfaces again in the present case. Recall that since the dimensionality of the spin chain was reduced at each step, it was necessary to account for this to obtain a monotonically increasing result. Similarly, we found that if the width of the network is monotonically reduced at subsequent layers, the relative entropy can decrease, and even become negative! While this seems to contradict the well-known proofs that the KL divergence is non-negative, the loophole is the assumption that both distributions are normalized with respect to the same measure. That is, consider the negative of the KL divergence between the one-dimensional distributions $p(x)$ and $q(x)$:

$$
\begin{aligned}
-D(p(x)\|q(x)) &= -\int \mathrm{d}x\, p(x)\ln\frac{p(x)}{q(x)} = \int \mathrm{d}x\, p(x)\ln\frac{q(x)}{p(x)} \\
&\leq \int \mathrm{d}x\, p(x)\left[\frac{q(x)}{p(x)}-1\right] = \int \mathrm{d}x\,[q(x)-p(x)] = 1-1 = 0\,,
\end{aligned}
\tag{91}
$$

where in going to the second line, we have used the fact that $\ln a \leq a-1$. In the present case however, we have $\int \mathrm{d}x\, p(x) = 1$ and $\int \mathrm{d}y\, q(y) = 1$ with $y = f(x)$, so that

$$
-D(p(x)\|q(y)) \leq \int \mathrm{d}x\, q(y) - 1 = \int \mathrm{d}y\left(\frac{\mathrm{d}y}{\mathrm{d}x}\right)^{-1} q(y) - 1\,,
\tag{92}
$$

and the "Jacobian" $\mathrm{d}y/\mathrm{d}x$ can be such that the right-hand side is greater than zero. In the present case, if the number of neurons changes from layer to layer, we do not have a true Jacobian, but rather a non-square matrix of first derivatives $\mathrm{d}z_i^\ell/\mathrm{d}z_j^{\ell-1}$. In principle, one can work out the corresponding change in the integration measure, implement it in code, and hope to amend the apparently pathological behavior. In practice however, for our purposes this is exceedingly complicated, and we can instead simply side-step the problem by maintaining the width of the network at a constant 784 neurons for MNIST, or 1024 for CIFAR-10[16]. We emphasize that while RG typically involves a reduction in the number of degrees of freedom, the hierarchical relationship underlying (1) is independent of dimensionality; in particular, we may take $\mathbf{x} = \{\mathbf{z}^\ell, \mathbf{z}^{\ell+1}\}$ for two layers of equal width $\mathbf{z}^\ell$, $\mathbf{z}^{\ell+1}$, and trace over $\mathbf{x}\backslash\mathbf{x}' = \mathbf{z}^\ell$ to obtain an effective distribution on $\mathbf{x}' = \mathbf{z}^{\ell+1}$. Holding the dimensions constant throughout the network thus affords a tremendous analytical simplification without fundamentally altering the structural relationship under study.

## 4 Discussion

In this work, we have taken a step towards quantifying information flow in lattice RG and deep neural networks, taking insight from both sides. Our main contribution is an explicit computation of the relative entropy or Kullback-Leibler divergence for both the 1d and 2d Ising models under decimation RG, as well as a simple feedforward random neural network of arbitrary depth. For the MNIST and CIFAR-10 datasets under study, the behavior is qualitatively identical in all cases[17]: we observe a relatively steep increase to some asymptotic value that depends on the choice of parameters (couplings and network initializations, respectively).

---

[16]That is, until the very last layer, which must be 10 neurons wide for the MNIST and CIFAR-10 classification tasks; in our experiments, we also reduced the penultimate layer to 400 neurons. Note that in order to train CIFAR-10 on our analytically tractable feedforward (non-covolutional) network, we converted the images to greyscale. For the reasons just explained, the KL divergence exhibits an unphysical drop in the last two layers due to the reduction in system size, which we have truncated from the plot in fig. 7.

[17]At least where the expressions are valid; as discussed in section 2, the approximation for the 2d Ising model breaks-down at sufficiently large coupling, resulting in an unphysical, non-monotonic decrease.

Stronger coupling in the spin system, or broader Gaussians for the weight matrices, results in larger values for the asymptote. For the Ising models, we are able to understand this in terms of the maximum information content in the system in the infinite-temperature limit, in which the spins decouple into $N$ non-interacting degrees of freedom, for an entropy of $N \ln 2$. Reducing the temperature – i.e., turning on interactions – cannot increase the entropy, so this represents the maximum amount of information we can possibly lose under RG.

For the deep neural networks, the interpretation of the value of the asymptote as a function of $\sigma_w^2$ is less clear. As discussed in section 3, the asymptotic value does not seem to correspond to the entropy one would naïvely assign to the reference (UV) layer, namely $\frac{1}{2} \ln |2\pi e \Sigma_{\ell=0}|$. Additionally, in fig. 7 we observe an increase in the value of the asymptote after training (but no qualitative change in the shape of the curve). This increase is relatively small and appears roughly constant (with respect to $\sigma_w^2$) for MNIST, but is larger and more varied for CIFAR-10; it would be interesting to quantify this on a wider range of supervised learning tasks to disentangle the change in the information content under training from that inherent in the structure of the network itself. Another potential direction would be to examine the KL divergence for different network architectures such as convolutional neural networks (CNNs), which seek to preserve spatial information within layers, and thus opens the possibility of quantifying the relative entropy of different subregions.

A related open question in machine learning is in disentangling compression and generalization in deep networks. To that end, one direction for future work is to investigate whether the asymptotic behavior imposes a fundamental bound on generalizability. That is, the process of deep learning – as distinct from the structure of the network – is very much like an RG in the sense that it involves a loss of information. Borrowing physics terminology, the whole point is to remove irrelevant (UV) information, and preserve only the relevant (IR) information necessary to compute the observables (e.g., perform the classification task) at hand. Once the KL divergence reaches its asymptotic value however, the total information content in the system no longer changes under subsequent layers / decimation steps—the RG is essentially completed. Intuitively, once too much information about the training data is removed, one might expect that the network will be unable to generalize; for example, some of the correlations in the training data may be common to the entire category of 5's or cats. Provided one could reliably disentangle the contributions to the relative entropy from the structure of the network vs. from the data, it would be interesting to explore whether training should be halted before the asymptotic value is reached. See [32, 42] for some entropic work in this vein, as well as [62] and references therein for connections to gradient descent and the neural tangent kernel.

More generally, as mentioned in the introduction, one can define the mutual information (MI) in terms of the KL divergence, and hence the asymptotic behavior may have implications for various algorithms that maximize MI, such as information flow maximization [42], or in bounding the MI in hidden representations in the information bottleneck [37, 38].

While an initial motivation for this work was to explore the notion of criticality through an entropic lens, we found that the KL divergence appears insensitive to the phase structure of both systems, beyond a monotonic increase in the asymptotic value as one moves further and further in the chaotic direction of parameter space. We believe the reason for this is that relative entropy is simply not well adapted to probing the structure of correlations that characterizes these phases. While [23, 24] and related works examined the two-point function of individual neurons, which exhibits the characteristic divergence in the correlation length we expect from physics, the KL divergence is only sensitive to the collective behavior of the entire system/layer. In the Ising models, this amounted to an expectation value of the Hamiltonian, which is determined by the coupling constants $K, K_0$; in the neural network, since we worked in the large-$N$ or mean-field limit, each layer was described by a Gaussian, which is entirely

characterized by its second cumulant (i.e., for $\mu = 0$, just $\sigma^2$). Thus, while the value of the asymptote itself may have some implication for network initialization or the study of lattice RG, the relative entropy does not appear to guide us in identifying the critical point itself. Nonetheless, given the impressive advantages in trainability observed for networks initialized near criticality, it may be of practical interest to better understand the "flow of information" in these systems in more precise, information-theoretic terms.

At a meta level, we present this work as a contribution to the rapidly growing intersection of physics and machine learning, whereby techniques from theoretical physics and information theory have been increasingly applied to the study of deep neural networks. For a small sample of other work on neural networks drawing from quantum and statistical field theory, in addition to those on criticality mentioned above, see for example [10, 28, 60, 63–65] and references therein.

## Acknowledgments

We thank James Giammona for many stimulating discussions, and for feedback on a draft of this manuscript. We also thank Dimitri Vvedensky for his course notes on the renormalization group. J.E. and K.T.G. acknowledge financial support from the Deutsche Forschungsgemeinschaft (DFG, German Research Foundation) under Germany's Excellence Strategy through the Würzburg-Dresden Cluster of Excellence on Complexity and Topology in Quantum Matter ct.qmat (EXC 2147, project id 390858490). K.T.G. also acknowledges the support of the Hallwachs-Röntgen Postdoc Program of ct.qmat.

## A  Decimation RG

To make this paper self-contained for the benefit of our machine learning readers, we will here review the real-space decimation RG procedure for the 1d classical Ising model. This is entirely standard and can be found in many textbooks; we shall here follow [53]. In section A.1, we have also included an explicit proof that this decimation procedure preserves the partition function.

For generality, let us include the external magnetic field $h$ (though this will be set to zero in the main text):

$$H = -J \sum_{\text{n.n.}} \sigma_i \sigma_j - h \sum_{i=1}^{N} \sigma_i \,, \tag{A.1}$$

with the periodic identification of boundaries $\sigma_{N+1} = \sigma_1$. For reasons which will shortly become apparent, we will also introduce a parameter $K_0$, so that the Hamiltonian becomes

$$H = -\sum_{i=1}^{N} \left[ K_0 + K_1 \sigma_i \sigma_{i+1} + \tfrac{1}{2} K_2 (\sigma_i + \sigma_{i+1}) \right] \,, \tag{A.2}$$

where $K_0 = 0$, $K_1 = \beta J$, and $K_2 = \beta h$; note that here we have also absorbed the inverse temperature into the couplings for notational convenience, so that the partition function for this model is given by

$$Z = \sum_{\{\sigma_i\}} e^{-H(\sigma_i)} \,, \tag{A.3}$$

where $\sum_{\{\sigma_i\}} := \prod_{i=1}^{N} \sum_{\sigma_i = \pm 1}$.

Now, each step of the RG flow corresponds to a decimation procedure in which we sum over half the degrees of freedom. Hence, taking $N$ even for simplicity, we wish to evaluate the

sum over $\sigma_i$ in (A.3) for which $i$ is even. To do so, we first re-express the exponential of the Hamiltonian (A.2) in the following form:

$$
\begin{aligned}
e^{-H(\sigma_i)} &= \prod_{i=1}^{N} \exp\left[K_0 + K_1 \sigma_i \sigma_{i+1} + \tfrac{1}{2}K_2(\sigma_i + \sigma_{i+1})\right] \\
&= \prod_{j=1}^{N/2} \exp\left[2K_0 + K_1(\sigma_{2j-1}\sigma_{2j} + \sigma_{2j}\sigma_{2j+1}) + \tfrac{1}{2}K_2(\sigma_{2j-1} + 2\sigma_{2j} + \sigma_{2j+1})\right],
\end{aligned}
\tag{A.4}
$$

whereupon the sum over all even spins becomes a sum over $\sigma_{2j} = \pm 1$, which yields

$$
\begin{aligned}
\sum_{\{\sigma_{2j}\}} e^{-H} &= \prod_{j=1}^{N/2} e^{2K_0} 2\cosh\left[K_1(\sigma_{2j-1} + \sigma_{2j+1}) + K_2\right] \exp\left[\tfrac{1}{2}K_2(\sigma_{2j-1} + \sigma_{2j+1})\right] \\
&= \prod_{j=1}^{N/2} e^{2K_0} 2\cosh\left[K_1(\sigma'_j + \sigma'_{j+1}) + K_2\right] \exp\left[\tfrac{1}{2}K_2(\sigma'_j + \sigma'_{j+1})\right],
\end{aligned}
\tag{A.5}
$$

where on the second line we have defined $\sigma'_j = \sigma_{2j-1} \implies \sigma'_{j+1} = \sigma_{2j+1}$. The partition function – that is, the sum over these remaining spins – can then be written

$$
Z = \sum_{\{\sigma'_j\}} \prod_{j=1}^{N/2} e^{2K_0} 2\cosh\left[K_1(\sigma'_j + \sigma'_{j+1}) + K_2\right] \exp\left\{\tfrac{1}{2}K_2(\sigma'_j + \sigma'_{j+1})\right\}.
\tag{A.6}
$$

The key step is then to express this partition function in the form (A.3), i.e., we define the renormalized couplings $K'_0$, $K'_1$, and $K'_2$ such that

$$
Z = \sum_{\{\sigma'_j\}} \exp\left\{\sum_{i=1}^{N/2}\left[K'_0 + K'_1 \sigma'_j \sigma'_{j+1} + \frac{1}{2}K'_2(\sigma'_j + \sigma'_{j+1})\right]\right\}.
\tag{A.7}
$$

In order for this to hold, we require that for all possible spin configurations,

$$
\begin{aligned}
&\exp\left[K'_0 + K'_1 \sigma'_j \sigma'_{j+1} + \frac{1}{2}K'_2(\sigma'_j + \sigma'_{j+1})\right] \\
&= e^{2K_0} 2\cosh\left[K_1(\sigma'_j + \sigma'_{j+1}) + K_2\right] \exp\left\{\tfrac{1}{2}K_2(\sigma'_j + \sigma'_{j+1})\right\}.
\end{aligned}
\tag{A.8}
$$

Configurations then fall into three equivalence classes: $\sigma'_j = \sigma'_{j+1} = \pm 1$, and $\sigma'_j = -\sigma_{j+1} = \pm 1$ (these last being equivalent in $Z$). Substituting these into (A.8) and solving for the renormalized coefficients then leads to the recursion relations

$$
\begin{aligned}
e^{K'_0} &= 2e^{2K_0}\left[\cosh(2K_1 + K_2)\cosh(2K_1 - K_2)\cosh^2 K_2\right]^{1/4}, \\
e^{K'_1} &= \left[\cosh(2K_1 + K_2)\cosh(2K_1 - K_2)\cosh^{-2} K_2\right]^{1/4}, \\
e^{K'_2} &= e^{K_2}\left[\cosh(2K_1 + K_2)/\cosh(2K_1 - K_2)\right]^{1/2}.
\end{aligned}
\tag{A.9}
$$

Note that even if $K_0$ were set to zero, a non-zero $K'_0$ would appear as a consequence of the change in normalization. Setting the external magnetic field $h$ (i.e., $K_2$) to zero, we obtain the recursion relations (13).

## A.1 Preservation of the partition function

Though the preservation of the partition function follows immediately from the Bayesian marginalization procedure discussed in the introduction, we can also show this explicitly for the real-space decimation prescription in the 1d Ising model. Recall from (15) that the logarithm of the partition function is given by

$$\ln Z = N \ln 2 + N K_0 + N \ln \cosh K. \tag{A.10}$$

Let $Z^{(n)}$ be the partition function after $n$ decimations:

$$\ln Z^{(n)} = \frac{N}{2^n} \ln 2 + \frac{N}{2^n} K_0^{(n)} + \frac{N}{2^n} \ln \cosh K^{(n)}. \tag{A.11}$$

Now, plug in the expression for $K_0^{(n)}$ in (14):

$$\ln Z^{(n)} = \frac{N}{2^n} \ln 2 + N \left(1 - \frac{1}{2^n}\right) \ln 2 + N K_0 + N \sum_{m=1}^{n} \frac{K^{(m)}}{2^m} + \frac{N}{2^n} \ln \cosh K^{(n)}$$

$$= N \ln 2 + N K_0 + N \sum_{m=1}^{n} \frac{K^{(m)}}{2^m} + \frac{N}{2^n} \ln \cosh K^{(n)}. \tag{A.12}$$

The statement we now wish to verify is that $Z^{(n)} = Z$ for all $n \in \mathbb{N}$, i.e., the partition function is preserved under decimation. Comparing (A.10) and (A.12), we see that we must have

$$\sum_{m=1}^{n} \frac{K^{(m)}}{2^m} + \frac{1}{2^n} \ln \cosh K^{(n)} = \ln \cosh K. \tag{A.13}$$

This is not at all obvious from the recursion relations (14), but it is in fact true and we can prove it using induction. The identity is trivial for $n = 0$. To prove the inductive step, we first find the following nontrivial identity:

$$\begin{aligned}
\frac{1}{2} \ln \cosh K^{(n+1)} &= \frac{1}{2} \ln \cosh \left( \frac{1}{2} \ln \cosh(2K^{(n)}) \right) \\
&= \frac{1}{2} \ln \frac{e^{\frac{1}{2} \ln \cosh(2K^{(n)})} + e^{-\frac{1}{2} \ln \cosh(2K^{(n)})}}{2} \\
&= \frac{1}{2} \ln \frac{\cosh^{\frac{1}{2}}(2K^{(n)}) + \operatorname{sech}^{\frac{1}{2}}(2K^{(n)})}{2} \\
&= \frac{1}{2} \ln \frac{\cosh(2K^{(n)}) + 1}{2 \cosh^{\frac{1}{2}}(2K^{(n)})} \\
&= \frac{1}{2} \ln \frac{\cosh^2(K^{(n)})}{\cosh^{\frac{1}{2}}(2K^{(n)})} \\
&= \ln \cosh K^{(n)} - \frac{1}{4} \ln \cosh(2K^{(n)}) \\
&= \ln \cosh K^{(n)} - \frac{K^{(n+1)}}{2}, \tag{A.14}
\end{aligned}$$

or

$$\frac{K^{(n+1)}}{2} + \frac{1}{2} \ln \cosh K^{(n+1)} = \ln \cosh K^{(n)}. \tag{A.15}$$

Note that plugging $n = 0$ into (A.15) gives precisely (A.13) with $n = 1$. The inductive step is then straightforward:

$$
\begin{aligned}
\sum_{m=1}^{n+1} \frac{K^{(m)}}{2^m} + \frac{1}{2^{n+1}} \ln \cosh K^{(n+1)} &= \sum_{m=1}^{n} \frac{K^{(m)}}{2^m} + \frac{K^{(n+1)}}{2^{n+1}} + \frac{1}{2^{n+1}} \ln \cosh K^{(n+1)} \\
&= \ln \cosh K - \frac{1}{2^n} \ln \cosh K^{(n)} + \frac{K^{(n+1)}}{2^{n+1}} + \frac{1}{2^{n+1}} \ln \cosh K^{(n+1)} \\
&= \ln \cosh K,
\end{aligned}
\tag{A.16}
$$

where the second line follows from the inductive hypothesis that (A.13) hold up to $n$, and the third line follows from (A.15). Thus, we have proven that $Z$ remains unchanged under decimation.

## B  Monte Carlo integrals

Here we include some basic details about the Monte Carlo (MC) integration used in computing the KL divergence for the random neural networks in section 3. MC methods are of course extremely well known: in general, one wishes to compute a multidimensional integral over some subregion $\Omega$ of $\mathbb{R}^d$,

$$
I = \int_{\Omega} d\mathbf{x}\, f(\mathbf{x}), \qquad \mathbf{x} \in \mathbb{R}^d,
\tag{B.1}
$$

where the volume of the integration region is

$$
V = \int_{\Omega} d\mathbf{x}.
\tag{B.2}
$$

Given the high-dimensional ($d = 784, 1024$) nature of the problem at hand, MC methods are inordinately more efficient than standard numerical integration, e.g., on some regular grid. Instead, the basic idea is to sample $n_{\mathrm{MC}}$ points $\mathbf{x}_i$, $i \in \{1, \ldots, n_{\mathrm{MC}}\}$ uniformly over $\Omega$. The law of large numbers then ensures that in the $n_{\mathrm{MC}} \to \infty$ limit, the integral will be given by the expectation value of the function $f(\mathbf{x})$ in the sample ensemble, i.e.,

$$
I \approx V \langle f \rangle \equiv \frac{V}{n_{\mathrm{MC}}} \sum_{i=1}^{n_{\mathrm{MC}}} f(\mathbf{x}_i).
\tag{B.3}
$$

However, this naïve algorithm is problematic in the present case, since the volume of the integration region is infinite, and the average value of the integrand is zero. This issue can be overcome via *importance sampling*, whereby, rather than drawing samples $\mathbf{x}_i$ uniformly, we draw them from a Gaussian $p(\mathbf{x}_i)$ with the same mean and standard deviation as the Gaussian integral under consideration. That is, our integrands take the form (writing the expressions in 1d for simplicity)

$$
f(x_i) = g(x_i) \frac{1}{\sigma \sqrt{2\pi}} e^{-\frac{1}{2}\left(\frac{x_i - \mu}{\sigma}\right)^2},
\tag{B.4}
$$

where $g(x_i)$ is some non-Gaussian factor. If we then draw samples from $p(x_i) = f(x_i)/g(x_i)$, then we have simply

$$
I \approx \frac{1}{n_{\mathrm{MC}}} \sum_i \frac{f(x_i)}{p(x_i)} = \frac{1}{n_{\mathrm{MC}}} \sum_i g(x_i).
\tag{B.5}
$$

Intuitively, the idea is that we weight each sample by an amount proportional to how often it appears in the ensemble, hence the name.

Let us now connect this to our algorithm in section 3. In the first step, we draw $n_{\text{MC}}$ samples of $\mathbf{z}^0 = \{z_i^0 \sim \mathcal{N}(\mu_0, \sigma_0^2)\}$, where $\mu_0$, $\sigma_0$ are the mean and standard deviation of the reference layer, respectively, and apply tanh to obtain ($n_{\text{MC}}$ samples of) $\phi(\mathbf{z}^0)$. Let us add an index $s$ to label the sample, i.e., $\phi_s(\mathbf{z}^0)$ with $s \in \{1, \ldots, n_{\text{MC}}\}$. The expectation value $f_j^{m-1}$ (85) is then obtained by simply averaging the $n_{\text{MC}}$ samples of the $j^{\text{th}}$ element of the vector of activations. That is, denoting the Gaussian measure on the reference layer by $\mathcal{D}\mathbf{z}^0$,

$$f_j^0 = \int \mathcal{D}\mathbf{z}^0 \, \phi(z_j^0) \approx \frac{1}{n_{\text{MC}}} \sum_{s=1}^{n_{\text{MC}}} \phi_s(z_j^0). \tag{B.6}$$

Similarly, to obtain $f_{j,k}^0$ (86), we simply plug in all possible products of $j, k$, taking care not to mix different samples when computing the averages:

$$f_{j,k}^0 = \int \mathcal{D}\mathbf{z}^0 \, \phi_s(z_j^0)\phi_s(z_k^0) \approx \frac{1}{n_{\text{MC}}} \sum_{s=1}^{n_{\text{MC}}} \phi_s(z_j^0)\phi_s(z_k^0). \tag{B.7}$$

The computation of the MC integrals (B.6), (B.7) is the same for all higher layers $m > 1$, except that we no longer need to perform any sampling: the integrals are always evaluated with respect to the reference layer, so we simply construct each sample of $\phi_s(z_i^m)$ from a sample of the previous layer $\phi_s(\mathbf{z}^{m-1})$, i.e.,

$$\phi_s(z_i^m) = \tanh\!\left(W_{ij}^m \phi_s(z_j^{m-1}) + b_i^m\right), \qquad m > 1, \tag{B.8}$$

and feed the collection of samples into the MC integrators to obtain $f_j^{m-1}, f_{j,k}^{m-1}$.

The code for the deep neural network analysis in section 3 and the computation described in this appendix is available here: https://github.com/ro-jefferson/entropy_dnn.

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
