# Peer review of "Towards quantifying information flows: relative entropy in deep neural networks and the renormalization group"

_SciPost Physics Core, doi:SciPost Phys. 12, 041 (2022)_

## Round 1 · Referee Report · Anonymous (Referee 1) · 2021-10-19

Strengths

The idea to compare quantitatively the KL-divergence between layers in neural networks and in statistical physics systems at different RG steps is very interesting. This comparison is nicely presented with natural starting examples on both sides. Many calculational details are provided which helps the reader to follow many steps.

Weaknesses

The main approach chosen in this article to study the relative entropy in deep neural networks is not well motivated. In particular, this work would benefit from a discussion on how the random network toy model is connected with the standard deep neural networks which are used in machine learning (i.e. with respect to training and different architectures).

The experiments on MNIST are too simple to draw meaningful empirical conclusions about the behaviour in deep networks. Simple linear multi-class classification already leads to relatively good results and the differences between different numbers of layers is not very pronounced in MNIST which is in contrast to other standard image classification tasks (e.g. on CIFAR-10 or CIFAR-100).

An empirical connection with trainability which goes beyond existing results in the literature is unclear. The current experiments are too weak to illustrate this connection.

Report

This article investigates the analogy between information processing in neural networks and the renormalization group flow in physical systems. This is a very timely article on a very interesting topic.

A short description on the strengths and weaknesses of the article are listed separately.

There are several changes which I suggest before recommending this article for publication.

Requested changes

1 - p2 Could the others clarify whether the analogy between variational RG is only for RBMs or does it apply more generally for deep Boltzmann machines? This is important to clarify the subsequent analogy with multiple layers in neural networks.

2 - From the current presentation, it is unclear how training of neural networks affects the comparison with RG and which parts of the analytical analysis need to be changed to explain these differences. I understand that a complete analysis is most likely beyond the scope of the article but a comment on where the empirically observed change can arise from would be beneficial for the article. This is also to understand how important results of random networks (i.e. networks before training) should be taken.

3 - The experiments on MNIST are too simplistic to allow for any meaningful conclusions about the behaviour in deep networks. This should be reflected in the discussion and conclusions. The current conclusions are too strong.

4 - A key aspect of calculating the KL-divergence is to take into account the different dimensions at different layers or respectively at different RG-steps. From the discussion at the end of section 3 it is unclear whether the previously adapted analytical procedure of dealing with different dimensionality is actually taken into account here. Could the authors clarify how the results represent the networks previously described with dimensional reduction. If there is an ambiguity on the layer dimension in the formalism, this should be reflected in the discussion.

---

## Round 1 · Referee Report · Anonymous (Referee 2) · 2021-11-1

Strengths

1)The authors suggest a rigorous way to construct a monotonic function to quantify the flow of information. Their numerical results are clear and presented in clear plots. Moreover, in the case of the Ising models the monotonicity has a physical interpretation.

2)Most of the calculations are presented in an analytic and clear way. Although some of them are fairly straightforward and maybe textbook material, they still add to the clarity and the good presentation of the paper.

3) The topic is very interesting and timely.

Weaknesses

1) The authors work with a relatively simple neural network and make a simple experiment to draw generic and strong conclusions.

2) The connection of the proposed function with the c-function is unclear. If the grounds for this claimed relation, it is only that both functions are monotonic then perhaps some sentences should be rephrased (e.g. in section 2). For example, the c-function and the entanglement entropy should be sensitive to the phase transitions while the monotonic function of the manuscript is not.

Report

The manuscript attempts to study the analogy between the renormalization group and the neural networks proposing a monotonic function based on relative entropy. The authors use the analytically solvable 1 and 2-dim Ising models, to introduce and establish their idea. Then they continue to apply them on a feedforward neural network. They find that the normalized function proposed exhibits a monotonic behaviour in the systems considered. Nevertheless, it is insensitive to the critical points and the phase diagram of the systems. This is a crucial difference with the monotonic c-function measuring the degrees of freedom in quantum field theory.

Requested changes

1) I find the results of the article interesting. However, I would suggest that the generic and strong conclusions of the article be scaled down. For example, before discussing the implications on the generalizability (e.g. mentioned in the abstract), it is more urgent/interesting to understand better the monotonic function proposed and how/if its properties hold in different neural network architectures and complex experiments.

2) I have a concern related to figure 4. I understand that the approximation breaks down for couplings above the critical one and this is the reason that the monotonicity changes. However, I would expect that numerically it would not be very difficult to compute the KL for couplings that are at least close to the critical one, especially for the small number of the RG steps considered. The authors should comment on this, and if there is no major numerical obstruction I would recommend the computation to show that the monotonicity is valid for any coupling.

3) Related to the previous point. Can the authors make any rigorous comments on their expectations of the function proposed on the Ising model with next-to-near neighbour interaction?

4) The connection with the c-function (if any), should be further clarified, otherwise some sentences should be rephrased(e.g. in section 2).

---

## Round 2 · Author Response

We thank both referees for their careful and reasonable reviews, and for raising several clarifying points which we have addressed below.

Response to Anonymous Report 1:

Here let us first respond to the general comments:

  • Concerning the use of random neural networks, these are chosen for analytical tractability, in keeping with the cited literature reviewed in section 3. As we discuss below, training does not fundamentally alter the analogy with RG, so the specific initialization scheme is qualitatively unimportant for our purposes. We have added a footnote (now footnote 13) in the text when introducing random networks about this, which we hope will also be more clear in light of our other edits.

  • Concerning the experiments, our empirical tests were intended to be merely preliminary, as the focus of this work is on establishing the mathematical parallel and exploring its theoretical potential. We have softened the strong statement in the conclusion to reflect the fact that we have only considered simple datasets, as the reviewer points out. At the same time, we have taken the reviewer's suggestion to additionally consider another standard classification task, namely CIFAR-10, which exhibits qualitatively identical results (see below).

  • Establishing an empirical connection to trainability was not the aim of this paper. As mentioned in the introduction (below eq. (1.1)), the RG captures interesting structural relationships, and it is these -- rather than training -- with which we are primarily concerned here. Indeed, one message of the paper belaboured in the introduction is that RG does not suffice to explain the dynamical process of learning (i.e., training), and in this sense we hope to constructively clarify certain exaggerated notions that have been put forth in the literature in this context.

Response to requested changes:

  1. As mentioned below eq. (1.1), this analogy holds for any hierarchical model. However, we implicitly used "RBM" to mean both traditional (i.e., two-layer) as well as stacked (i.e., deep) RBMs, which we believe may have caused the confusion. We have elaborated on this to make clear that it does not depend on the number of layers.

  2. As discussed below eq. (1.1) as well as on page 27/29, the analogy with RG holds at the level of structure rather than dynamics (i.e., training/learning). We have elaborate on the discussion on page 27/29 to further clarify the fact that the analytical analysis is unchanged, and also provide more intuition for the mentioned empirical change (namely, a quantitative but not qualitative shift). We have also added a comment on the role of initialization in this context.

  3. We agree with the reviewer that the experiments are very simplistic, and have strengthened this by repeating the analysis for CIFAR-10 (converted to greyscale and trained on our simple feedforward network, where the computation of the KL divergence is explicitly tractable), which exhibits the same qualitative behaviour, thereby demonstrating that this is not specific to the dataset. Additionally, we have softened the conclusion to make clear that we have only considered these two datasets (to further highlight the potential interest in extending this to a wider range of supervised learning tasks), as well as commented on extensions to different architectures such as CNNs.

  4. At the end of the mentioned discussion about different dimensions in section 3, we state explicitly that we hold the network width constant for our experiments to avoid the complicated task of computing the change in the integration measure, so there is no ambiguity to contend with. Nonetheless, in the hopes of clarifying the previously discussed relationship with RG, we have added a comment at the very end of this section, explaining that the constant dimension does not alter the analogy.

Response to Anonymous Report 2:

Concerning the general comments: we have addressed the simplicity of our experiments in contrast to our conclusions above, and will address the question of the c-function in the itemised responses below.

Requested changes:

  1. This overlaps with comments by the previous referee; see our response regarding the softening of our conclusions and strengthening of our analysis above. Additionally, regarding the mention of "generalizability" in the Discussion, we have presented this as an interesting direction for future study, not as a conclusion that follows from our work. Similarly, the relationship between generalizability and trainability is well-known in the machine learning literature. Nevertheless, we have softened the language used when reviewing this idea for the reader.

  2. In this case, the problem is not a numerical difficulty per se, but that the analytical approximation itself breaks down at strong-coupling. In order to compute the RG, even numerically, we must write down a recursion relation for the couplings, but the approximation (2.43) is justified only a posteriori based on momentum-space (i.e., exact) results. Real-space RG does not yield consistent recursion relations at strong-coupling; this is a well-known problem in the literature, as mentioned in footnote 11 and below eq. (2.44). We have added a new paragraph at the top of page 14 (including footnote 12) to further elaborate this issue in the case of the 2d Ising model.

  3. Insofar as the decimation RG effectively induces next-to-nearest neighbour interactions after marginalising over UV degrees of freedom (and that eventually, we will have diluted the interactions to the point where there is no additional information to remove), we would expect qualitatively similar behaviour, but we do not have a particularly rigorous comment about this. Mathematically of course, the KL divergence must be non-decreasing, though the approach to the asymptote may differ.

  4. Throughout this paper, we refer to the c-theorem in the broader sense that there is some function counting the number of degrees of freedom that decreases along RG flows, as made explicit by the relation between the c-function and entropy that we refer to on page 2. We certainly do not claim that the relative entropy we have identified is equal to the c-function in the Zamolodchikov sense in any perturbed CFT, asymptoting to the central charge at the fixed points. For this reason, the fact that one is insensitive to the phase behaviour is immaterial to this connection. Nonetheless, we see how our statements could be confusing to the reader, and we have added a footnote on page 9 (footnote 9) to make this clear.

We hope that you will kindly consider the resubmitted manuscript for publication in SciPost.

Sincerely yours J. Erdmenger, K. Grosvenor, and R. Jefferson

---

## Round 2 · List of Changes

For convenience, here we collect a list of the changes above:

1. Added comment below eq. (1.1) clarifying the applicability of the RG analogy to deep RBMs.
2. Added references [34,35] on page 3.
3. Added footnotes 9, 12, and 13
4. Corrected typo (join --> joint) on page 12.
5. Repeated computation of the KL divergence for CIFAR-10; added additional plot in fig. 7 with results, and updated various mentions of our experiments in the text accordingly.
6. Repeated fitting of the asymptote for CIFAR-10; added additional plot in fig. 8 and results for parameters in Table 1.
7. Elaborated on the role of structure vs. dynamics (and initialization) on pages 27/29.
8. Elaborated on the dimensional dependence/normalization at the end of section 3.
9. Softened the conclusion, and incorporated new CIFAR-10 results into the discussion.
10. Added comments about future directions (e.g., CNNs) in the discussion.

---

## Editorial Decision

published